# A multi-ethnic epigenome-wide association study of leukocyte DNA methylation and blood lipids

Min-A Jhun [1,2 ✉], Michael Mendelson [3,4], Rory Wilson [5,6], Rahul Gondalia[7], Roby Joehanes [8], Elias Salfati[2], Xiaoping Zhao[9], Kim Valeska Emilie Braun [10], Anh Nguyet Do[11], Åsa K. Hedman [12], Tao Zhang[13], Elena Carnero-Montoro[14,15], Jincheng Shen[16], Traci M. Bartz[17], Jennifer A. Brody [18], May E. Montasser[19,20], Jeff R. O'Connell[19,20], Chen Yao [3], Rui Xia[9], Eric Boerwinkle[21], Megan Grove[21], Weihua Guan[22], Pfeiffer Liliane[5,6], Paula Singmann [5,6], Martina Müller-Nurasyid [23,24,25], Thomas Meitinger[25,26,27], Christian Gieger[5,6], Annette Peters [6,25], Wei Zhao [1], Erin B. Ware [1,28], Jennifer A. Smith [1,28], Klodian Dhana[29], Joyce van Meurs[30], Andre Uitterlinden [30], Mohammad Arfan Ikram [10], Mohsen Ghanbari [10], Deugi Zhi[31], Stefan Gustafsson [12], Lars Lind[12], Shengxu Li[13], Dianjianyi Sun[13,32], Tim D. Spector [14], Yii-der Ida Chen[33], Coleen Damcott [19,20], Alan R. Shuldiner[19,20], Devin M. Absher[34], Steve Horvath [35,36], Philip S. Tsao [2,37,38], Sharon Kardia[1], Bruce M. Psaty[39,40], Nona Sotoodehnia[41], Jordana T. Bell [14], Erik Ingelsson[2,12,38,42], Wei Chen [13], Abbas Dehghan[10,43], Donna K. Arnett [11], Melanie Waldenberger [5,6], Lifang Hou[44], Eric A. Whitsel[7,45], Andrea Baccarelli[46,47], Daniel Levy[3,48], Myriam Fornage [9], Marguerite R. Irvin[11] & Themistocles L. Assimes [2,37,38 ✉]

Here we examine the association between DNA methylation in circulating leukocytes and blood lipids in a multi-ethnic sample of 16,265 subjects. We identify 148, 35, and 4 novel associations among Europeans, African Americans, and Hispanics, respectively, and an additional 186 novel associations through a trans-ethnic meta-analysis. We observe a high concordance in the direction of effects across racial/ethnic groups, a high correlation of effect sizes between high-density lipoprotein and triglycerides, a modest overlap of associations with epigenome-wide association studies of other cardio-metabolic traits, and a largely non-overlap with lipid loci identified to date through genome-wide association studies. Thirty CpGs reached significance in at least 2 racial/ethnic groups including 7 that showed association with the expression of an annotated gene. CpGs annotated to *CPT1A* showed evidence of being influenced by triglycerides levels. DNA methylation levels of circulating leukocytes show robust and consistent association with blood lipid levels across multiple racial/ethnic groups.

A full list of author affiliations appears at the end of the paper.

Abnormal blood lipid levels are important risk factors for various diseases including cardiovascular disease[1], diabetes[2], renal disease[3], Alzheimer's disease[4], and cancers[5,6]. Several large-scale genome-wide[7,8] and exome-wide association studies of lipids[9,10] have identified single nucleotide polymorphisms (SNPs) involved in lipid metabolism. However, the biological mechanisms behind abnormalities in lipid metabolism are not fully understood.

Complex traits are a manifestation of not only genetic but also environmental factors which in part express themselves through the epigenetic modification of DNA in all cell types. Epigenetic modifications can explain differences in a phenotype between monozygotic twins[11] as well as changes in a phenotype within an individual over time[12]. Epigenome-wide association studies (EWAS) provide an opportunity to document differences in epigenetic marks between individuals through the quantification of the degree of methylation at thousands of CpG sites across the genome. To date, such studies have identified methylation signatures associated with a number of cardiometabolic traits including cigarette smoking[13], BMI[14], hepatic fat[15], fasting insulin or HOMA-IR[16], incident type2 diabetes[17], renal function[18], blood pressure[19], and C-reactive protein[20]. In addition, several EWAS identified CpGs significantly associated with blood lipid levels in populations of European ancestry[21–24]. However, large-scale multi-ethic studies to identify epigenetic determinants of blood lipid levels are lacking[25].

Here, we present a racial/ethnic groups specific meta-analysis of 15 EWAS of DNA from circulating white blood cells involving a total 16,265 participants from 3 racial/ethnic groups to identify CpGs with DNA methylation levels that are significantly associated with blood lipid levels. We identify 187 racial/ethnic specific novel CpG associations among Europeans, African Americans, and Hispanics and an additional 186 novel association through a trans-ethnic meta-analysis. To aid in the interpretation of our results, we quantify the consistency of associations across racial/ethnic groups, determine overlap between our findings and previously published relevant genome wide and epigenome-wide association studies, explore for the presence of cis-methylation quantitative trait loci (cis-mQTL) and/or cis-expression quantitative trait methylation (cis-eQTM) for CpG associations found to be significant in at least 2 racial/ethnic groups, and attempt to provide evidence on the direction of these associations using bi-directional Mendelian randomization.

## Results

**Study population.** We analyzed 12 cohorts of Europeans (EA) involving 11,114 participants, 7 cohorts of African Americans (AA) involving 4,425 participants, and 2 cohorts of Hispanics (HISP) involving 699 participants (Table 1, Supplementary methods). The TwinsUK, WHI-BA23, and WHI-EMPC cohorts were composed of female participants only while NAS was composed of male participants only. The range of mean age, body mass index (BMI), high-density lipoprotein (HDL) levels, low-density lipoprotein (LDL) levels, and triglyceride (TG) levels was 42.7 to 76.0 year, 26.6 to 32.6 kg per m$^2$, 45.5 to 59.3 mg per dl, 104.9 to 152.6 mg per dl, 74.1 to 168.5 mg per dl, respectively (Table 1). The percentage of study participants taking any lipid control medication at time of blood lipid measurement ranged between 0% in the Amish to 44% in FHS.

**EWAS stratified by racial/ethnic group.** We measured DNA methylation levels using the Illumina Infinium HumanMethylation 450 K Beadchip in peripheral blood leukocytes or whole blood, except in GOLDN where CD4 + T cells exclusively were used (Supplementary Data 1). We performed an EWAS on HDL, LDL, and TG using four linear mixed effects models in each cohort, stratified by racial/ethnic group and a random effects meta-analysis[26] with genomic control (GC) and Bonferroni correction for the number of probes tested ($P < 1.09 \times 10^{-7}$) (Methods, Supplementary Table 1).

**Table 1 Descriptive of the participated cohorts ($N = 16,265$).**

| Cohort | N | Age (year) Mean ± SD | Female Percent | BMI (kg/m$^2$) Mean ± SD | Percent on Lipid meds | HDL (mg/dl) Mean ± SD | LDL (mg/dl) Mean ± SD | TG (mg/dl) Mean ± SD |
|---|---|---|---|---|---|---|---|---|
| European ($N = 11,114$) | | | | | | | | |
| Amish | 158 | 43.8 ± 13.2 | 54 | 27.8 ± 4.7 | 0 | 55.8 ± 14.9 | 139.5 ± 39.9 | 74.1 ± 39.4 |
| BHS | 676 | 42.8 ± 4.5 | 55 | 29.9 ± 6.7 | 12 | 45.5 ± 13.2 | 126.7 ± 34.2 | 142.4 ± 92.9 |
| CHS | 186 | 76.0 ± 5.0 | 55 | 27.1 ± 5.0 | 5 | 51.7 ± 13.5 | 117.1 ± 36.4 | 155.8 ± 82.0 |
| FHS* | 2,648 | 66.4 ± 8.9 | 54 | 28.3 ± 5.3 | 44 | 57.3 ± 18.0 | 104.6 ± 31.4 | 118.9 ± 69.8 |
| GOLDN* | 714 | 48.5 ± 15.9 | 50 | 28.5 ± 5.5 | 0 | 46.1 ± 13.0 | 123.8 ± 31.5 | 137.4 ± 85.8 |
| KORA F4 | 1,651 | 61.0 ± 8.9 | 51 | 28.1 ± 4.8 | 16 | 56.5 ± 14.7 | 140.3 ± 35.3 | 129.9 ± 77.8 |
| NAS | 674 | 72.4 ± 6.8 | 0 | 28.0 ± 4.1 | 36 | 49.3 ± 12.9 | 120.1 ± 33.0 | 137.7 ± 84.0 |
| PIVUS | 963 | 70.1 ± 0.15 | 50 | 27.0 ± 4.3 | 16 | 58.6 ± 16.4 | 128.9 ± 33.9 | 112.6 ± 51.4 |
| RS | 724 | 59.9 ± 8.2 | 54 | 27.5 ± 4.8 | 26 | 54.3 ± 15.9 | 134.7 ± 37.6 | 131.5 ± 76.0 |
| TwinsUK* | 708 | 58.1 ± 9.3 | 100 | 26.6 ± 4.8 | 16 | 71.3 ± 17.8 | 123.9 ± 38.9 | 99.5 ± 52.1 |
| WHI-BA23 | 940 | 68.3 ± 6.25 | 100 | 28.8 ± 5.9 | 14 | 51.0 ± 12.3 | 142.7 ± 37.6 | 154.0 ± 35.5 |
| WHI-EMPC | 1,072 | 64.7 ± 7.1 | 100 | 28.8 ± 5.8 | 10 | 59.2 ± 15.8 | 133.0 ± 34.1 | 158.6 ± 80.1 |
| African ($N = 4,452$) | | | | | | | | |
| ARIC | 1,877 | 56.5 ± 5.9 | 64 | 30.0 ± 6.2 | 8 | 53.4 ± 17.0 | 134.7 ± 38.8 | 112.1 ± 57.3 |
| BHS | 282 | 42.7 ± 4.7 | 61 | 32.6 ± 8.7 | 7 | 49.6 ± 14.5 | 120.8 ± 33.9 | 114.8 ± 76.9 |
| CHS | 189 | 73.0 ± 5.5 | 67 | 28.8 ± 5.0 | 4 | 59.3 ± 16.5 | 124.2 ± 34.7 | 114.6 ± 58.6 |
| GENOA* | 315 | 61.1 ± 7.6 | 73 | 31.0 ± 6.3 | 6 | 56.9 ± 17.4 | 125.2 ± 41.4 | 136.1 ± 63.6 |
| HyperGEN | 604 | 48.4 ± 11.2 | 67 | 32.5 ± 8.2 | 5 | 53.6 ± 15.0 | 122.6 ± 38.0 | 107.8 ± 62.5 |
| WHI-BA23 | 652 | 62.8 ± 6.6 | 100 | 31.9 ± 6.8 | 17 | 54.8 ± 14.6 | 152.6 ± 43.8 | 122.6 ± 81.3 |
| WHI-EMPC | 533 | 62.7 ± 6.9 | 100 | 31.5 ± 6.1 | 10 | 58.1 ± 14.7 | 135.9 ± 37.9 | 119.6 ± 54.3 |
| Hispanic ($N = 699$) | | | | | | | | |
| WHI-BA23 | 389 | 62.2 ± 6.9 | 100 | 29.6 ± 5.4 | 16 | 50.6 ± 13.2 | 142.7 ± 37.6 | 168.5 ± 95.6 |
| WHI-EMPC | 310 | 61.6 ± 6.2 | 100 | 29.6 ± 5.3 | 11 | 54.7 ± 12.9 | 128.8 ± 35.1 | 162.7 ± 77.1 |

*A family-based cohort; N Number of samples, SD Standard deviation, HDL high-density lipoprotein, LDL low-density lipoprotein, TG triglycerides.

We identified 447, 25, and 496 CpGs for HDL, LDL, and TG, respectively, among EA using the basic set of covariates (model 1 adjusted for age, sex, smoking, lipid medication, four SNP PCs, estimated cell proportions, plate, row, and column of plate) (Methods, Supplementary Data 2). When we further adjusted for BMI (model 2 additionally adjusted for BMI), the numbers of significant CpGs decreased substantially for HDL (146) and TG (206) but increased modestly for LDL (30). Among AA, we identified 34, 7, and 76 CpGs in model 1 for HDL, LDL, and TG, respectively, and the numbers decreased to 9, 7, and 55 with a further adjustment with BMI (model 2). For HISP, we identified 2, 0, and 6 CpGs in model 1 for HDL, LDL, and TG, respectively, and the number decreased to 0 for HDL. Excluding participants taking any lipid lowering medication decreased the sample size by 18% and decreased power but the effect estimates remained similar (models 3 and 4 adjusted for the same set of covariates of models 1 and 2, respectively, with the exception of adjustment for the use of lipid medications, Methods). Among EA, we identified 74, 15, and 86 CpGs significantly associated ($P < 1.09 \times 10^{-7}$) with HDL, LDL, and TG, respectively, using this most conservative model 4 that excluded statin users and adjusted for BMI (Fig. 1, Supplementary Data 2, Supplementary Fig. 1). For AA, these numbers were 7, 5, and 43 and, for HISP, they were 2, 2, and 4 CpGs, respectively. Through trans-ethnic meta-analyses of the same model, we additionally identified 49, 24, and 119 significant ($P < 1.09 \times 10^{-7}$) CpG-lipid level associations for HDL, LDL, and TG, respectively, of which 46, 22, and 118 were novel when compared to both our racial/ethnic specific analyses and the literature.

**Comparison of results across racial/ethnic groups.** A majority of significant CpG-lipid level associations in European population did not reach statistical significance in other racial/ethnic groups. For CpGs found to be significant in at least one racial/ethnic group (model 4), we found a high rate of concordance (88 to 100%) in the direction of effect observed in Europeans versus that observed in African Americans and, separately, Hispanics for all three lipid fractions (Fig. 2). We found a high correlation between effect sizes (Pearson correlation coefficient = 0.69 to 0.93) for HDL and TG but not for LDL (−0.20 to 0.20). Lastly, regression slopes between any 2 sets of betas were close to 1 for HDL (0.75, 1.03) and TG (0.68 to 1.12) but not for LDL (−0.28 to 0.26) (Fig. 2). We calculated the correlations coefficients and estimated regression slopes for a higher number of CpGs (122 CpGs with $P < 1.09 \times 10^{-5}$) before and after natural log transformation to further explore whether the differences in correlations and regression slopes between LDL and HDL/TG could be a consequence of having much smaller number of significant CpGs and/or the use of a log transformed lipid measure. For these analyses, we found both higher correlation of betas (0.21 to 0.49) and regression slopes (0.34 to 0.47) for LDL although still not as high as those observed for HDL and TG (Supplementary Fig. 2).

When comparing results across all 3 racial/ethnic groups, we identified 4, 1, and 26 CpGs associated with HDL, LDL, and TG, respectively, in more than one racial/ethnic group (Table 2). Of these, 1 CpG, cg06500161, in *ABCG1* was associated with both HDL and TG in opposing directions. Consistent with our findings for significant CpG-lipid trait associations overall, we found high rate of concordance of the direction of the associations of the 30 CpGs across all 3 racial/ethnic group but variable effect sizes (Supplementary Figs. 3-5). The 4 CpGs in the *ABCG1*, carnitine palmitoyltransferase 1 A (*CPT1A*), sterol regulatory element binding transcription factor 1 (*SREBF1*), and thioredoxin interacting protein (*TXNIP*) genes identified in HISP for TG

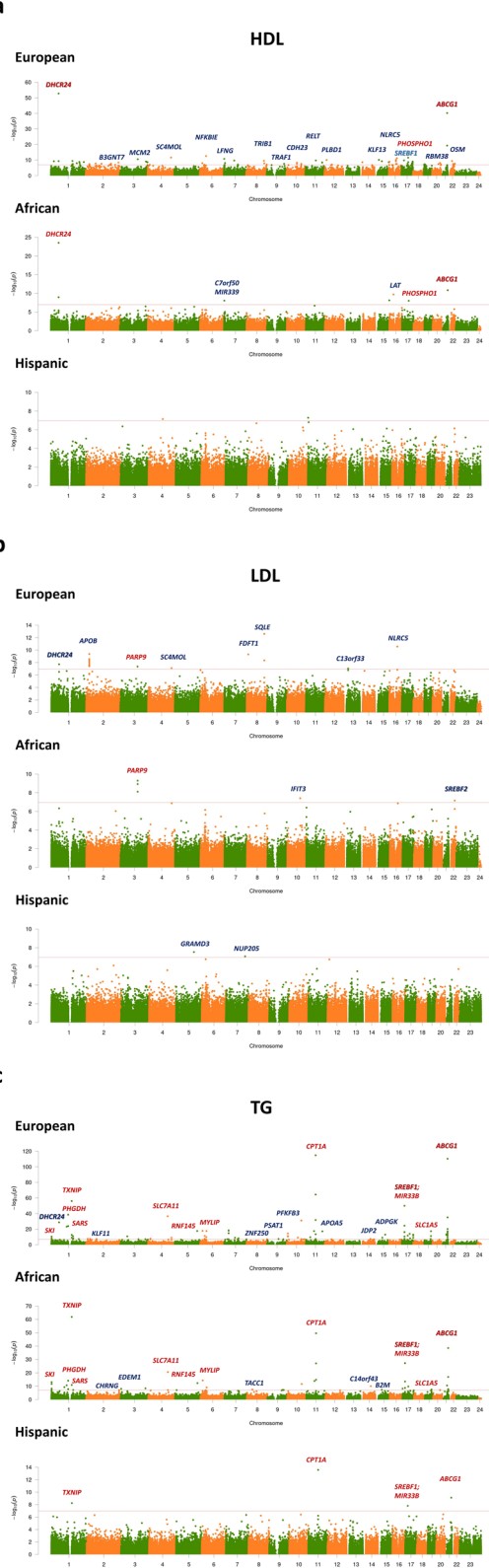

were also significant in the EA and AA populations. We found multiple CpGs to be significantly associated with TG levels located within or near the phosphoglycerate dehydrogenase (*PHGDH*), *CPT1A*, *SREBF1*, and *ABCG1* genes. Twelve out of the 30 replicated CpGs significant in at least 2 racial/ethnic groups have not been previously reported[21–24] (Table 2).

**Fig. 1 Manhattan plots for the meta-analyses of the epigenome-wide association studies.** Manhattan plots for (**a**) high-density lipoprotein (HDL), (**b**) low-density lipoprotein (LDL), and (**c**) triglycerides (TG) in European ($N = 11{,}114$), African ($N = 4{,}452$), and Hispanic ($N = 699$) populations. Results are plotted as negative log-transformed $P$ values (y-axis) across the genome (x-axis). Odd chromosomes are in green and even chromosomes are in orange. The red horizontal line represents the epigenome-wide significance threshold of $1.09 \times 10^{-7}$. Linear mixed effects models were implemented adjusting for age, sex (reference = male), smoking variable (never/previous/current, reference = never), lipid medication (Yes or No, reference = No), the top four principal components from genotypes (SNPs), the proportion of 5 types of cells estimated with the Houseman method (CD8 T lymphocytes, CD4 T lymphocytes, natural killer cells, B cells, and monocytes), and random effects for plate, row, and column, and BMI (model 4). The top CpGs of each chromosome were annotated with a gene name (in blue font: identified in a racial/ethnic group; red: identified in multiple racial/ethnic groups; bold: significantly associated with multiple lipid measures).

**Overlap with prior related GWAS and EWAS.** We examined whether the CpGs identified in our lipid EWAS are located in or near the genes implicated in previous lipid GWAS[7,27,28]. Among EA, a total of 4 out of 106 genes had both CpGs and SNPs significantly associated with a blood lipid: apolipoprotein A5 (*APOA5*), apolipoprotein B (*APOB*), myosin regulatory light chain interacting protein (*MYLIP*), and *PARP9* (Fig. 3). Among AA, 4 out of 32 genes had been identified in both lipid EWAS and GWAS: *PARP9*, Chromosome 7 Open Reading Frame 50 (*C7orf50*), ATP Binding Cassette Subfamily A Member 1 (*ABCA1*), and *MYLIP*. We further identified 192 CpG-SNP pairs within 10Mbp at an additional 19 loci of which 30 pairs at 10 loci were within 1Mbp (Supplementary Data 5).

Many of the CpGs identified in our lipid EWAS have been previously identified in EWAS of BMI[14], hepatic fat[15], fasting insulin or HOMA-IR[16], incident type2 diabetes[17], estimated glomerular filtration rate (eGFR)[18], blood pressure[19], C-reactive protein[20], and/or cigarette smoking[13] (Fig. 3, Supplementary Table 2). About one half of the genes ($33/192 = 17\%$ of CpGs) identified in our BMI adjusted lipid EWAS have also been implicated in previously published BMI EWAS[14]. There were also twelve genes identified in more than two EWAS: *ABCG1*, *CPT1A*, 24-dehydrocholesterol reductase (*DHCR24*), *PHGDH*, phosphoethanolamine/phosphocholine phosphatase (*PHOSPHO1*), seryl-tRNA synthetase (*SARS*), SKI proto-oncogene (*SKI*), solute carrier family 1 member 5 (*SLC1A5*), solute carrier family 43 member 1 (*SLC43A1*), *SLC7A11*, *SREBF1*, and *TXNIP*. Among EA, 37 out of 106 genes (39 out of 164 CpGs) have been identified in EWAS of other phenotypes. There were 14 out of 32 genes (14 out of 54 CpGs) and 2 out of 6 genes (4 out of 8 CpGs) have been identified in EWAS of other phenotypes for AA and HISP, respectively.

**Methylation quantitative trait loci analysis.** We searched for methylation quantitative trait loci (mQTL) influencing methylation levels of the 30 CpGs listed in Table 2. Five out of the 15 cohorts provided genetic data for this analysis including ARIC ($N_{AA} = 1{,}717$), GOLDN ($N_{EA} = 713$), KORA ($N_{EA} = 1{,}379$), WHI-BA23 ($N_{EA} = 790$, $N_{AA} = 540$, and $N_{HISP} = 324$), and WHI-EMPC ($N_{EA} = 494$, $N_{AA} = 424$, and $N_{HISP} = 221$). We restricted our analysis to SNPs located within 25 kilobases up- or downstream of the CpGs with a minor allele frequency (MAF) > 0.01 in each cohort and implemented a fixed effects meta-analysis within each of the three racial/ethnic groups. A total of 11, 18, and 5 CpGs had at least one significant mQTL in EA (number of

tests = 5549, Bonferroni corrected $P = 9.01 \times 10^{-6}$), AA (number of tests=8316, Bonferroni corrected $P = 6.01 \times 10^{-6}$), and HISP (number of tests=4,713, Bonferroni corrected $P = 1.06 \times 10^{-5}$), respectively (Supplementary Data 3). We found 7 out of our 11 CpGs in EA to also be listed to have at least one mQTLs in the mQTL DB (http://www.mqtldb.org/)[29] (Supplementary Data 4). For the 7 CpGs, 95% of the significant mQTL SNPs in mQTL DB were also significant in our EA population and had consistent direction of effect. Out of the 190 significant mQTLs (SNP-CpG pairs) common to both our study and the mQTL DB, 51 (27%) were found to be mQTLs in datasets spanning across the life course in the mQTL DB including birth, childhood, adolescence, pregnancy, and middle age.

**Expression quantitative trait methylation analysis.** The association between DNA methylation and gene expression was investigated in the Framingham Heart Study ($N_{EA} = 4{,}278$ including 2,726 offspring cohort participants and 1552 third generation cohort participants)[30]. The DNA methylation levels of 7 out of the 30 CpGs listed in Table 2 were negatively associated with the expression of their respective genes: phosphoglycerate dehydrogenase (*PHGDH*) (cg14476101: $P = 4.09 \times 10^{-9}$), poly (ADP-ribose) polymerase family member 9 (*PARP9*) (cg22930808 with a transcript Chr3:122398047–122449684: $P = 5.58 \times 10^{-10}$; with a transcript Chr3:122246779–122283503: $P = 3.27 \times 10^{-6}$), solute carrier family 7 member 11 (*SLC7A11*) (cg06690548: $P = 3.34 \times 10^{-12}$), *CPT1A* (cg09737197: $P = 2.67 \times 10^{-10}$; cg17058475: $P = 1.18 \times 10^{-11}$), *ABCG1* (cg07397296: $P = 6.26 \times 10^{-8}$; cg06500161: $P = 4.44 \times 10^{-53}$). Three of these CpGs are in the 5' UTR region while the remaining four are in the gene body. Aside from one CpG in a CpG island, six were located in either the north or south shore regions (Supplementary Table 3).

**Mendelian randomization approach.** We explored the causal relationships between methylation and blood lipid levels for the 30 CpGs in EA (Table 2) using a bi-directional Mendelian Randomization (MR) study design[31,32]. First, we used genetic risk scores (GRS) for HDL, LDL, and TG constructed from established susceptibility loci for these phenotypes as instruments to examine the relationship between blood lipids and methylation (Supplementary Table 4). We found the GRSs to be significantly associated with their respective lipid levels in the 4 cohorts (GOLDN ($N_{EA} = 713$), KORA ($N_{EA} = 1{,}379$), WHI-BA23 ($N_{EA} = 790$), and WHI-EMPC ($N_{EA} = 494$) participating in the MR follow-up analysis (HDL: $P_{meta} = 1.86 \times 10^{-37}$, LDL: $P_{meta} = 1.13 \times 10^{-22}$, TG: $P_{meta} = 1.13 \times 10^{-8}$). Our Mendelian randomization analysis suggested that the DNA methylation levels of three CpGs, cg00574958 ($P = 4.23 \times 10^{-6}$), cg17058475 ($P = 4.72 \times 10^{-4}$), and cg09737197 ($P = 3.33 \times 10^{-3}$), located in the 5'UTR region of the *CPT1A* were influenced by blood TG levels (Supplementary Table 5). Next, we investigated the effect of DNA methylation on blood lipid levels. A total of 7 out of the 30 CpGs had at least one significant mQTL with available GWAS results from the Global Lipids Genetics Consortium results (Supplementary Table 6). We implemented inverse-weighted MR method and MR-egger when >2 mQTLs were available for a given CpG (3 CpGs out of 7). None of the CpGs were significantly associated with lipid levels using MR-egger.

**Discussion**

We report the first large-scale multi-ethnic epigenome-wide association study (EWAS) of blood lipids. Our population specific meta-analyses revealed 187 novel CpG-lipid trait associations and identified a high concordance of the direction of effects across racial/ethnic groups for all 3 lipid traits and a high correlation of

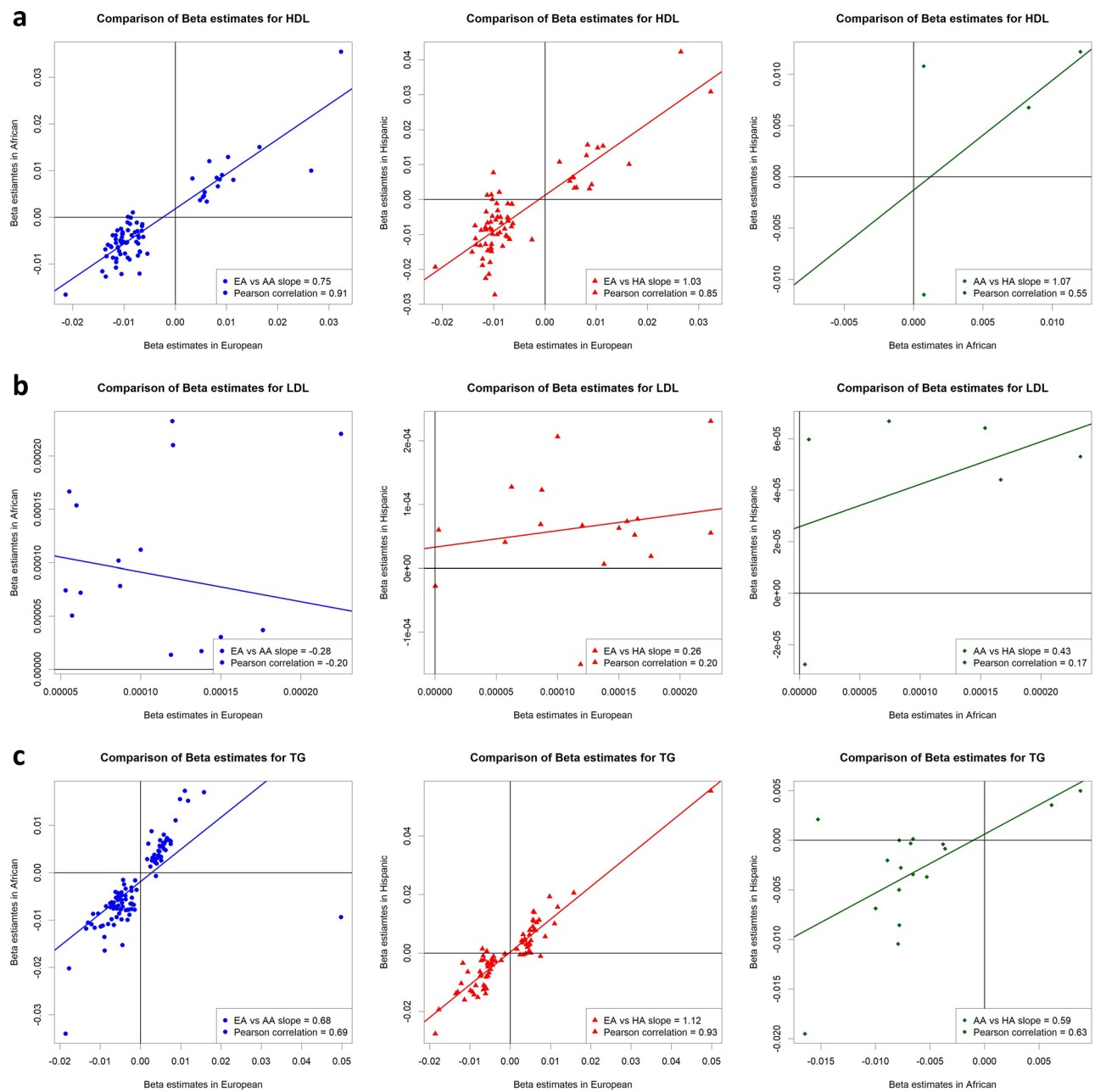

**Fig. 2 Scatter plots and regression lines of beta estimate pairs observed in two racial/ethnic groups for CpG-lipid trait associations.** Plotted are betas from CpGs found to be significant ($P < 1.09 \times 10^{-7}$) in one or more racial/ethnic group. Plots and lines are shown between Europeans (EA) and African Americans (AA) (blue arrowheads/lines), Europeans (EA) and Hispanics (HA) (red triangles/lines), and African Americans (AA) and Hispanics (HA) (green diamonds/lines) for (**a**) high-density lipoprotein (HDL), (**b**) low-density lipoprotein (LDL), and (**c**) triglycerides (TG). Numerical regression slope and Pearson correlation coefficients are presented in the bottom right corner of each plot.

effects sizes for associations with HDL and TG. A majority of our significant CpG-lipid associations do not implicate genes previously identified through GWAS of lipids[7–10,27,28], but many of our associations overlap with those identified in EWAS to date of related cardiometabolic traits especially for TG and HDL[14–19,33]. Thirty CpG-lipid trait associations were significant in at least 2 racial/ethnic groups with ~1/3 of these being novel. In subgroup analyses, 19 significant CpGs also harbored mQTLs and 7 were inversely associated with levels of expression of the annotated gene. Lastly, our Mendelian randomization analyses suggested that DNA methylation levels at one locus appeared to be influenced by blood TG levels.

The numbers of statistically significant CpGs decreased dramatically for HDL and TG after adjusting for BMI. These findings suggest that BMI serves as either a strong confounder or a strong mediator of a large fraction of our CpG-lipid associations for these traits. Even after adjusting for BMI, we found ~1/3 of CpGs to be linked to BMI in other studies[14]. In addition to the potential of residual confounding, we hypothesize that many CpGs may be independently influenced by both BMI and blood lipid levels (akin to how diet and exercise have an independent effect on weight loss).

We found a relatively high overlap of findings from our study with previous EWAS findings for cardiometabolic traits and a

**Table 2 CpGs significantly associated with lipids levels in more than one racial/ethnic group.**

| CpG | RACE | BETA | SE | P | MAPINFO | GENE | GROUP | CpG Island | Enhancer | DHS |
|---|---|---|---|---|---|---|---|---|---|---|
| **HDL** | | | | | | | | | | |
| cg17901584* | EA | 0.032 | 0.005 | $1.68 \times 10^{-53}$ | 1:55353706 | DHCR24 | TSS1500 | S_Shore | | |
| | AA | 0.036 | 0.003 | $3.05 \times 10^{-24}$ | | | | | | |
| cg24002003 | EA | 0.010 | 0.002 | $2.39 \times 10^{-09}$ | 15:101668143 | | | | Y | Y |
| | AA | 0.013 | 0.002 | $7.32 \times 10^{-09}$ | | | | | | |
| cg02650017* | EA | 0.006 | 0.001 | $2.82 \times 10^{-12}$ | 17:47301614 | PHOSPHO1 | Body | Island | Y | |
| | AA | 0.005 | 0.001 | $9.70 \times 10^{-09}$ | | | | | | |
| cg06500161* | EA | −0.021 | 0.003 | $6.32 \times 10^{-41}$ | 21:43656587 | ABCG1 | Body | S_Shore | Y | |
| | AA | −0.017 | 0.002 | $1.45 \times 10^{-11}$ | | | | | | |
| **LDL** | | | | | | | | | | |
| cg22930808 | EA | 0.0001 | 0.00002 | $4.38 \times 10^{-08}$ | 3:122281881 | PARP9 | 5'UTR | N_Shore | | |
| | AA | 0.0002 | 0.00003 | $1.18 \times 10^{-09}$ | | | | | | |
| **TG** | | | | | | | | | | |
| cg03725309 | EA | −0.007 | 0.0011 | $6.29 \times 10^{-24}$ | 1:109757585 | SARS | Body | S_Shore | | Y |
| | AA | −0.008 | 0.001 | $2.21 \times 10^{-11}$ | | | | | | |
| cg16246545 | EA | −0.011 | 0.002 | $1.33 \times 10^{-24}$ | 1:120255941 | PHGDH | Body | S_Shore | | |
| | AA | −0.012 | 0.0019 | $1.48 \times 10^{-08}$ | | | | | | |
| cg14476101* | EA | −0.018 | 0.003 | $2.88 \times 10^{-39}$ | 1:120255992 | PHGDH | Body | S_Shore | | |
| | AA | −0.020 | 0.0024 | $7.53 \times 10^{-15}$ | | | | | | |
| cg19693031* | EA | −0.019 | 0.0021 | $9.14 \times 10^{-57}$ | 1:145441552 | TXNIP | 3'UTR | | | |
| | AA | −0.034 | 0.0034 | $1.31 \times 10^{-62}$ | | | | | | |
| | HA | −0.028 | 0.0132 | $5.76 \times 10^{-9}$ | | | | | | |
| cg19266329 | EA | −0.006 | 0.0008 | $2.62 \times 10^{-13}$ | 1:145456128 | | | | Y | |
| | AA | −0.009 | 0.0013 | $1.17 \times 10^{-11}$ | | | | | | |
| cg19213703 | EA | −0.005 | 0.0012 | $9.42 \times 10^{-08}$ | 3:177554561 | | | | Y | |
| | AA | −0.008 | 0.0019 | $4.39 \times 10^{-09}$ | | | | | | |
| cg06690548* | EA | −0.012 | 0.0024 | $2.92 \times 10^{-37}$ | 4:139162808 | SLC7A11 | Body | | | |
| | AA | −0.011 | 0.0011 | $2.71 \times 10^{-21}$ | | | | | | |
| cg26403843 | EA | 0.011 | 0.002 | $2.21 \times 10^{-18}$ | 5:158634085 | RNF145 | Body | N_Shelf | | |
| | AA | 0.017 | 0.0022 | $5.12 \times 10^{-13}$ | | | | | | |
| cg03717755* | EA | 0.009 | 0.001 | $1.56 \times 10^{-18}$ | 6:16136539 | MYLIP | Body | | Y | |
| | AA | 0.011 | 0.0017 | $4.87 \times 10^{-15}$ | | | | | | |
| cg18336453 | EA | −0.004 | 0.0008 | $2.31 \times 10^{-10}$ | 6:43082296 | PTK7 | Body | | Y | |
| | AA | −0.007 | 0.0011 | $8.87 \times 10^{-10}$ | | | | | | |
| cg07504977* | EA | 0.012 | 0.0016 | $8.24 \times 10^{-32}$ | 10:102131012 | | | N_Shelf | Y | |
| | AA | 0.015 | 0.002 | $2.47 \times 10^{-12}$ | | | | | | |
| cg11376147* | EA | −0.004 | 0.0006 | $3.54 \times 10^{-18}$ | 11:57261198 | SLC43A1 | Body | | Y | Y |
| | AA | −0.006 | 0.001 | $1.32 \times 10^{-14}$ | | | | | | |
| cg00574958* | EA | −0.009 | 0.0012 | $1.50 \times 10^{-115}$ | 11:68607622 | CPT1A | 5'UTR | N_Shore | | |
| | AA | −0.011 | 0.0011 | $2.34 \times 10^{-50}$ | | | | | | |
| | HA | −0.013 | 0.0025 | $2.65 \times 10^{-14}$ | | | | | | |
| cg09737197* | EA | −0.008 | 0.0013 | $1.86 \times 10^{-32}$ | 11:68607675 | CPT1A | 5'UTR | N_Shore | | |
| | AA | −0.011 | 0.002 | $1.38 \times 10^{-15}$ | | | | | | |
| cg17058475* | EA | −0.010 | 0.0013 | $4.29 \times 10^{-65}$ | 11:68607737 | CPT1A | 5'UTR | N_Shore | | |
| | AA | −0.011 | 0.0025 | $8.52 \times 10^{-28}$ | | | | | | |
| cg10919522 | EA | −0.005 | 0.001 | $5.59 \times 10^{-08}$ | 14:74227441 | C14orf43 | 5'UTR | S_Shore | Y | |
| | AA | −0.010 | 0.0014 | $1.02 \times 10^{-10}$ | | | | | | |

**Table 2 (continued)**

| CpG | RACE | BETA | SE | P | MAPINFO | GENE | GROUP | CpG Island | Enhancer | DHS |
|---|---|---|---|---|---|---|---|---|---|---|
| cg15863539* | EA | 0.003 | 0.0009 | $8.83 \times 10^{-17}$ | 17:17716950 | SREBF1 | Body | S_Shore | | |
| | AA | 0.004 | 0.0006 | $1.93 \times 10^{-09}$ | | | | | | |
| cg20544516* | EA | 0.006 | 0.0006 | $3.71 \times 10^{-25}$ | 17:17717183 | MIR33B;SREBF1 | Body;Body | S_Shore | | |
| | AA | 0.007 | 0.0008 | $3.18 \times 10^{-14}$ | | | | | | |
| cg11024682* | EA | 0.010 | 0.0013 | $1.08 \times 10^{-50}$ | 17:17730094 | SREBF1 | Body | S_Shelf | | |
| | AA | 0.016 | 0.0013 | $5.77 \times 10^{-28}$ | | | | | | |
| | HA | 0.019 | 0.0033 | $1.59 \times 10^{-8}$ | | | | | | |
| cg08857797* | EA | 0.006 | 0.001 | $1.93 \times 10^{-12}$ | 17:40927699 | VPS25 | Body | | | |
| | AA | 0.008 | 0.0017 | $1.73 \times 10^{-10}$ | | | | | | |
| cg22304262 | EA | −0.006 | 0.0012 | $6.04 \times 10^{-12}$ | 19:47287778 | SLC1A5 | Body;5'UTR | N_Shelf | | |
| | AA | −0.008 | 0.0012 | $3.31 \times 10^{-08}$ | | | | | | |
| cg08309687 | EA | −0.009 | 0.0017 | $4.49 \times 10^{-14}$ | 21:35320596 | | | | Y | Y |
| | AA | −0.014 | 0.0024 | $3.00 \times 10^{-11}$ | | | | | | |
| cg0188189* | EA | 0.005 | 0.0008 | $6.07 \times 10^{-21}$ | 21:43652704 | ABCG1 | Body | N_Shelf | | |
| | AA | 0.007 | 0.0011 | $1.21 \times 10^{-17}$ | | | | | | |
| cg02370100* | EA | 0.005 | 0.0008 | $1.45 \times 10^{-13}$ | 21:43655256 | ABCG1 | Body | Island | | |
| | AA | 0.006 | 0.0014 | $1.49 \times 10^{-08}$ | | | | | | |
| cg07397296 | EA | 0.007 | 0.0012 | $4.68 \times 10^{-16}$ | 21:43655316 | ABCG1 | Body | Island | | Y |
| | AA | 0.007 | 0.0012 | $3.98 \times 10^{-08}$ | | | | | | |
| cg06500161* | EA | 0.016 | 0.0015 | $4.73 \times 10^{-111}$ | 21:43656587 | ABCG1 | Body | S_Shore | Y | |
| | AA | 0.017 | 0.0023 | $2.72 \times 10^{-39}$ | | | | | | |
| | HA | 0.021 | 0.0033 | $7.68 \times 10^{-10}$ | | | | | | |

*Identified in previous epigenome-wide association of lipids; SD Standard error, CHR Chromosome, DHS DNase I hypersensitive site, HDL high-density lipoprotein, LDL low-density lipoprotein, TG triglycerides, EA European ancestry, AA African ancestry, HA Hispanic ancestry, $N_{EA} = 11,114$, $N_{AA} = 4,452$, $N_{HA} = 699$; The statistic is a likelihood ratio test (LRT) which asymptotically follows an equal mixture of 1 degree of freedom (df) $\chi^2$ distribution and 2 df $\chi^2$ distribution; CI can be calculated as (BETA-1.96*SE, BETA +1.96*SE). A p-value was considered significant in a specific racial/ethnic group if it was <$1.09 \times 10^{-7}$ (Bonferroni correction for the number of CpG probes tested).

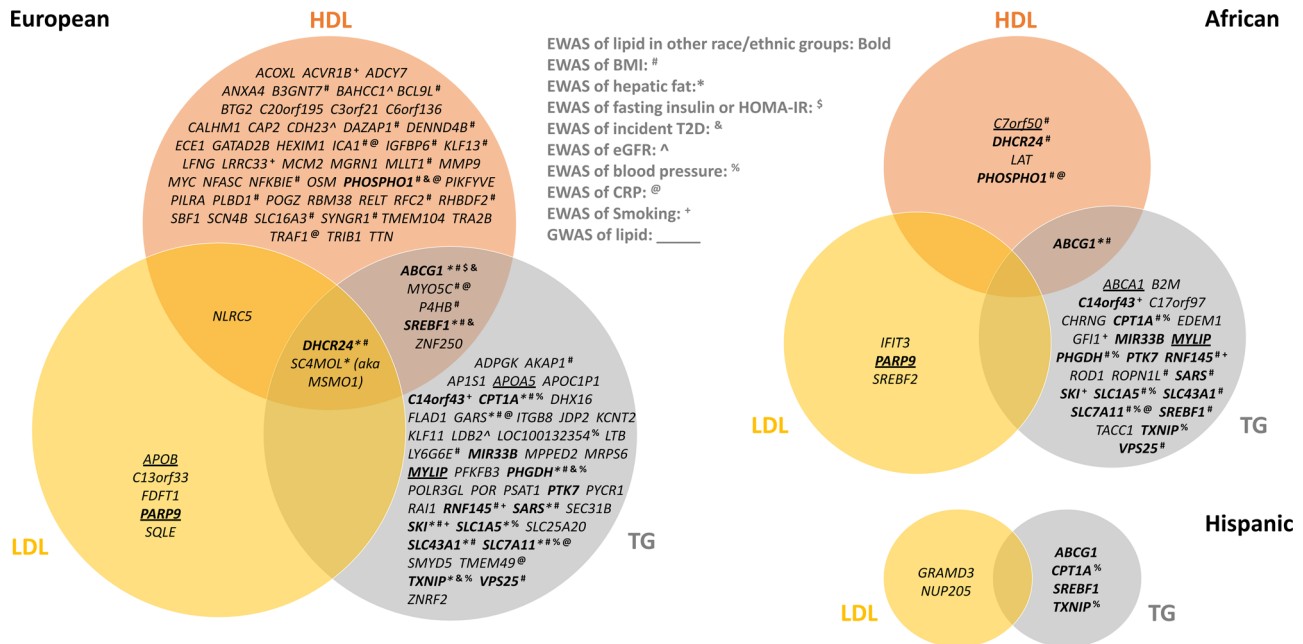

**Fig. 3 Venn diagram of genes identified through epigenome-wide association study of lipids and their overlap with other EWAS as well as genome-wide association studies of lipids.** Genes identified in the racial/ethnic specific (European, African, and Hispanic) stratified meta-analysis of CpG-lipid associations involving either high-density lipoprotein (HDL), low-density lipoprotein (LDL), or triglycerides (TG) levels after adjusting for BMI and excluding subjects on statins. Genes identified in more than one racial/ethnic population are in bold font. Genes identified in EWAS of related cardiometabolic traits are marked with special characters: BMI (#), hepatic fat (*), fasting insulin or HOMA-IR ($), incident type2 diabetes (&), eGFR (^), blood pressure (%), C-reactive protein (CRP) (@), and cigarette smoking (+). Genes identified in previous GWAS are underlined.

relative paucity of overlap with lipid loci implicated through GWAS. We hypothesize that the influence of important upstream environmental determinants of the metabolic syndrome such as diet and exercise may be responsible for these patterns although substantial additional research is needed to prove this hypothesis including studies measuring changes in epigenetic profiles of multiple relevant cell types after dietary and physical activity interventions.

We could not detect methylation QTLs (mQTLs) for many of our 30 CpGs-lipid trait associations replicating across 2 racial/ethnic groups. Among mQTLs identified, a majority were not consistently associated with the CpG over the life course[29]. Whether this observation reflects direct changes in methylation levels of CpGs associated with lipid levels that occur over a lifetime of accumulated environmental exposures, such as diet and exercise, remains unknown. Other environmental exposures or time-dependent events leading to subtle changes in white blood cell proportions may also be responsible for these observations.

We highlight 3 EWAS lipid loci uncovered through our meta-analysis among the plethora of both novel and known findings. First, we found a CpG to be associated with *PARP9* among our few LDL findings. *PARP9* is homologous to *PARP10* and both are ADP-ribosyltransferases with 30% of their catalytic domains being identical[34,35]. Both also have been previously identified as either LDL or total cholesterol loci through GWAS[28]. Second, we identified CpG associations in *ABCG1* for both HDL and TG. *ABCG1* is a member of the superfamily of ATP-binding cassette transporters that plays an important role in macrophage cholesterol efflux. Notably, *ABCA1*, another member of this gene family, has also been robustly linked to the control of HDL and TC levels through GWAS[36]. We replicate the previously reported association between *ABCG1* methylation and *ABCG1* expression[37] and note that expression levels of *ABCG1* have also been found to be positively correlated with obesity[38]. Lastly, the methylation status of cg06500161 in *ABCG1* is associated with an elevated risk of

developing type 2 diabetes[39] while genetic variants mapped to this locus are linked to atherosclerosis[40,41]. These constellations of findings suggest that *ABCG1* may play a role in predisposing to or mediating the effects of the metabolic syndrome. Third, we found the methylation status of CpGs in *CPT1A*, a gene that initiates the oxidation of long-chain fatty acids, to be influenced by blood levels of TG through our MR analysis. The same CpG (cg00574958) in *CPT1A* was also found to be influenced by blood pressure levels in another EWAS[19]. In other human and animal studies, DNA methylation levels of the CpGs in *CPT1A* have been associated with *CPT1A* expression[21], plasma adiponectin levels[42], the metabolic syndrome[43], BMI[14], hepatic fat[15], and high fructose consumption[44,45]. Collectively, these findings suggest that methylation status of *CPT1A* may mediate the downstream effects of the metabolic syndrome.

The two major strengths of our study are its size and ethnic diversity. These strengths allowed for improved power to detect novel CpG-blood lipid trait associations and to robustly explore the generalizability of the findings across multiple racial/ethnic groups. Our study has limitations in other respects some of which are common to all EWAS studies of blood. First, it remains unclear whether epigenetic changes in blood cells serve as a good surrogate of changes in the most relevant tissues controlling blood lipid levels[46]. To establish a reliable surrogate tissue, interindividual epigenetic differences must not only correlate between blood and the tissue of interest but also exposures must induce similar changes to both tissues[46]. Such evidence either does not yet exist or is incomplete for most trait-exposure combinations[47–50]. While circulating leucocytes are likely to exert at least partial direct control over blood lipid levels[51,52], other relevant tissues/cells that we did not examine include hepatocytes, adipocytes, and enterocytes. A second limitation common to all EWAS of circulating leucocytes includes the potential for findings to not be truly reflective of a chronic alteration of transcriptional regulation from environmental perturbations but rather residual

confounding due to persistent cell subtype proportional heterogeneity despite the application of statistical deconvolution techniques[46,53,54]. Lastly, the cross-sectional design of our study makes it difficult to determine the directionality of our associations[54]. We attempted to provide additional evidence of directionality using Mendelian randomization techniques[32], but the power of our MR analyses was limited due to the lack of availability of genetic data for many cohorts limiting sample size combined with a lack of strong genetic instruments for many of CpGs examined[55]. In addition, the known shared genetic background of HDL, LDL, and TG introduces the possibility of biases due to pleiotropy in our MR analysis.

In conclusion, we identified 373 novel CpG-lipid traits associations through the largest multi-ethnic EWAS to date. We found a high concordance of the direction of effects for all 3 lipids traits across racial/ethnic groups and a high correlation of effects for HDL and TG with 30 CpGs—including 12 novel CpGs—reaching stringent statistical significance in at least 2 racial/ethnic groups. A large majority of implicated genes do not overlap with lipid loci identified to date through GWAS although many loci associated with HDL and TG in >2 racial/ethnic groups have been associated with related cardio-metabolic traits in previous EWAS. We provide some limited insights on mechanism of association through our mQTL, eQTM, and MR analyses but additional studies are needed before firm conclusions can be made on the causality and mechanisms behind a large majority of the associations we observed.

## Methods

**Study populations**. A total of 15 cohorts ($N = 16,265$) from the epigenetics working group in the Cohorts for Heart and Aging Research in Genomic Epidemiology (CHARGE) consortium participated in this study. These included the Old Order Amish (OOA), Atherosclerosis Risk in Communities (ARIC), Bogalusa Heart Study (BHS), Cardiovascular Health Study (CHS), Framingham Heart Study (FHS), Genetic Epidemiology Network of Arteriopathy (GENOA), Genetics of Lipid Lowering Drugs and Diet Network (GOLDN), Hypertension Genetic Epidemiology Network (HyperGEN), Cooperative health research in the Region of Augsburg (KORA), Normative Aging Study (NAS), Prospective Investigation of Vascularity of Uppsala Elders Study (PIVUS), Rotterdam Study (RS), UK Adult Twin Registry (TwinsUK), Women's Health Initiative Broad Agency Announcement 23 (WHI-BA23), and the Women's Health Initiative Epigenetic Mechanisms of PM-Mediated CVD (WHI-EMPC) cohorts. Four cohorts, BHS, CHS, WHI-BA23, and WHI-EMPC, examined more than one racial/ethnic group. The total number of cohorts in the European, African, and Hispanic study populations is 12 ($N = 11,114$), 9 ($N = 4,452$), and 2 ($N = 699$), respectively. The participating cohorts are described in the Supplementary materials. All studies obtained written informed consent from participants and were approved by local institutional review boards and ethics committees.

**Lipid measurements**. High-density-lipoprotein (HDL, mg per dl) and triglycerides (TG, mg per dl) were directly measured in blood samples taken from participants after at least an 8 h fast. Low-density-lipoprotein (LDL, mg per dl) was inferred using the Friedewald's formula[56] in all cohorts except for GOLDN, HyperGEN, and KORA where LDL was measured directly. We did not infer LDL in subjects with triglycerides >400 mg per dL and we excluded lipid measure from subjects who did not fast for at least 8 h. We also excluded outliers as defined by >5 standard deviations from the mean of blood lipid in each cohort. To reduce skewness, HDL and triglycerides were natural log-transformed.

**DNA methylation measurement, QC, and normalization**. DNA methylation was produced by investigators from each cohort independently. Levels were measured from peripheral blood leukocytes isolated from whole blood in all studies except GOLDN where only CD4 + T cells were examined. The EZ DNA Methylation Gold Kit (Zymo Research, Orange CA) was used for bisulfite conversion. The Illumina® Infinium HumanMethylation450 BeadChip and the Illumina BeadXpress reader were used to perform the methylation assays. Either the SWAN[57] method in the *minfi*[58] R package, the Beta Mixture Quantile method (BMIQ)[59], the DASEN method in the wateRmelon R package[60], or the GenomeStudio® Methylation Module was used for pre-processing and normalization of the data in each cohort (Supplementary Data 1). For each CpG site, a beta-value was calculated representing the percent methylation at that CpG site. We used an annotation file provided on the Illumina website to annotate CpGs to genes. CpGs were annotated to genes by Illumina using the following rules: those located within 1500 bp

upstream of transcription start site (TSS1500), TSS200, 5′UTR, 1st exon, gene body, or 3′UTR of a gene were annotated to that gene. All other intergenic CpGs were not annotated to a gene. To reduce technical batch effects, plate, row, and column information were added as random effects in the association analyses. To reduce confounding from cellular heterogeneity[61], we estimated cell proportions using Houseman's method[53] in each subject and used these proportions as covariates in the association analyses.

Any single value with a detection *p*-value > 0.01 was set to missing. In each cohort, we excluded probes with a detection *p*-value > 0.01 in greater than 5% of samples. In addition, we excluded samples with a detection *p*-value > 0.01 in greater than 5% of probes. To avoid spurious signals in DNA methylation data, we excluded 29,233 CpGs that co-hybridize to alternate genomic sequences (highly homologous to the intended targets)[62].

**Epigenome-wide association study**. Epigenome-wide association analyses (EWAS) were performed in each cohort stratified by racial/ethnic group (European, African, and Hispanic). For Model 1, a linear mixed effects model was used to study the association between the DNA methylation level of a CpG (dependent variable) and each of the lipid measures (independent variable; HDL, LDL, or TG), adjusting for age, sex (reference = male), smoking variable (never/previous/current, reference = never), lipid medication (Yes or No, reference = No), the top four principal components from genotypes (SNPs), and the proportion of 5 types of cells estimated with the Houseman method (CD8 T lymphocytes, CD4 T lymphocytes, natural killer cells, B cells, and monocytes)[53]. We added random effects for plate, row, and column. We also included family structure as a random effect among family-based studies. For Model 2, we further adjusted for BMI in addition to Model 1 covariates. We also ran Model 3 and Model 4 which were analogous to Model 1 and 2, respectively, in the subset of individuals not taking lipid-lowering medication.

**Meta-analysis**. We performed meta-analyses of all the participating cohorts ($N = 16,265$) and also stratified by racial/ethnic group: European Americans (12 cohorts, $N = 11,114$), African Americans (7 cohorts, $N = 4,452$), and Hispanics (2 cohorts, $N = 699$). These meta-analyses were performed for each of the 4 models, respectively. We used a random effects meta-analysis implemented in METASOFT[26] to take into account the heterogeneity of the effect sizes of different cohorts while achieving a higher or comparable statistical power compared to fixed effects meta-analysis. To avoid spurious findings from population substructure, we applied genomic control[63]. We considered a $p < 1.09 \times 10^{-7}$ to be significant, equivalent to a Bonferroni correction for the number of CpG probes. We then compared results for all CpGs across all three racial/ethnic groups to identify the subset of CpGs with significant associations across two or more racial/ethnic groups. These CpGs were prioritized and followed-up with mQTL analysis, eQTM analysis, and a Mendelian randomization approach for causal inference.

**Overlap with prior related genome-wide association studies**. To identify the overlap between EWAS CpGs and genome-wide association studies (GWAS) SNPs of lipids, we tried three approaches: (1) Identify a GWAS SNP and a EWAS CpG pair located within 1 Mbp; (2) Identify a GWAS SNP and a EWAS CpG pair located within 10 Mbp; and (3) Identify a GWAS SNP and a EWAS CpG pair annotated to the same gene. SNPs were annotated to a gene if it is located within the transcript boundary of a protein-coding gene or a nearest gene if it is located outside of genes. The CpGs were annotated to a gene if it is located within 1500 base pairs of the transcription start site, 5′-UTR, gene body, or 3′-UTR using the Illumina annotation file. Intergenic CpGs were not compared for this 3rd approach.

**Methylation quantitative trait loci**. We investigated the association between either imputed or genotyped SNPs located within 25 kb upstream or downstream of each CpG and DNA methylation levels to identify cis-acting methylation quantitative trait loci (mQTL). For imputed SNP data, we restricted the mQTL analysis to SNPs with a good quality imputation (IMPUTE info > =0.4 or MACH r^2 > =0.3). Subjects taking lipid lowering medications were excluded from this analysis. Beta-values of DNA methylation levels were inverse-normal transformed and regressed on age, sex, smoking (current/former/never), BMI, at least 4 SNP PCs, cell proportions (WBC count and/or estimated WBC proportions (granulocytes as a reference)), and technical covariates (plate, row, and column as random effects) (two-sided test). Family information was also included as a random effect if a cohort was a family-based study. We then regressed the residuals on each SNP of interests stratified by racial/ethnic group.

**Expression quantitative trait methylation**. We extracted associations between DNA methylation levels and gene expression in blood (expression quantitative trait methylation or eQTM) from a pre-existing investigation involving 4278 participants of the FHS (2,726 offspring cohort participants and 1552 third-generation cohort participants)[30]. For gene expression, whole blood was collected in PAXgene™ tubes (PreAnalytiX, Hombrechtikon, Switzerland) and frozen at −80 °C. RNA was extracted using the whole blood RNA System Kit (Qiagen, Venlo, Netherlands) and mRNA expression profiling was assessed using the Affymetrix

Human Exon 1.0 ST GeneChip platform (Affymetrix Inc, Santa Clara, CA), which contains more than 5.5 million probes targeting the expression of 17,873 genes. The Robust Multi-array Average (RMA) package[64] as used to normalize the gene expression values and remove any technical or spurious background variation.

Linear mixed effects model was used to assess associations between residuals of DNA methylation levels and residuals of gene expression levels (two-sided test). We first regressed out the fixed effects of age, sex, white blood cell type proportions as estimated through the Houseman method[53], technical variables (sample storage time, RNA integrity number), and the first component of a principal component analysis and the random effects of amplification batch in the gene expression levels. Next, we regressed out the fixed effects of age, sex, Houseman's white blood cell type proportions, and the first two PCs of a principal component analysis and the random effect of batch (plate) from the DNA methylation levels. We then applied an additional adjustment for 25 surrogate variables to both the gene expression and DNA methylation data before the methylation-gene expression association analysis. Surrogate variables were calculated using the gene expression data (residualized for PC1, and technical variables as fixed effects and amplification batch as random effect) and the DNA methylation data (residualized for PC1 and PC2 as fixed effects and batch as random effect) adjusting for age, sex, and Houseman's blood cell type prediction.

**Mendelian randomization approach**. We applied a bi-directional Mendelian randomization approach to shed light on the causal nature of associations we identified between blood lipid levels and CpGs across ≥ 2 racial/ethnic groups[31,32,65–67]. First, we created genetic risk scores (GRS) for increasing lipid levels using either imputed (where IMPUTE info > =0.4 or MACH r^2 > =0.3) or genotyped SNPs known to be robustly associated with lipids through prior large scale GWAS[7,8]. The GRSs for each subject were calculated as the sum of the lipid (HDL, LDL, or TG) increasing allele dosages of the SNPs listed in Supplementary Table 4 divided by the number of SNPs used to calculate the specific GRS. We performed a Mendelian randomization analysis in each study among participants not taking lipid lowering medication followed by a meta-analysis of analogous results in each study. Each lipid measure was regressed on the corresponding GRS adjusting for age, sex, smoking, BMI, at least 4 SNP PCs, and a family effect if a cohort was a family-based study. The DNA methylation level of each CpG was also regressed on the corresponding GRS adjusting for age, sex, smoking, BMI, at least 4 SNP PCs, cell proportions, technical batch effects, and a family effect if a cohort was family-based study. The study-specific estimates and standard errors of the GRS-lipid and the GRS-CpG associations were used as input for a meta-analysis to obtain overall estimates for GRS-lipid and GRS-CpG associations. The overall estimates were then used to assess the causal effect of each lipid measure to a CpG with a Mendelian randomization approach.

Secondly, we implemented Mendelian randomization[65,66] method to study the effect of DNA methylation on blood lipids levels. The identified significant mQTL SNPs were used as an instrument for the CpGs of interest. Study participants taking any lipid lowering medication were excluded in the Mendelian randomization analysis. Beta-values of DNA methylation levels were inverse-normal transformed and regressed on age, sex, smoking (current, former, or never), BMI, at least 4 SNP PCs, cell proportions (WBC count and/or estimated WBC proportions (gran as a reference)), and technical covariates (plate, row, and column as random effects). Family information was also included as a random effect if a cohort is a family-based study. Then the residuals were regressed on each SNP located 25 kb up and downstream of the CpG site. The mQTL analyses were performed separately for each racial/ethnic group. Each SNPs were also searched to identify associations with lipid levels in the previous GWAS results from Global Lipids Genetics Consortium (GLGC, http://csg.sph.umich.edu/willer/public/lipids2013/)[7]. The estimates and standard errors of the SNP-CPG (obtained from our samples) and SNP-lipid associations (obtained from the GLGC GWAS results) were used as input for inverse-weighted MR and MR-egger methods to assess the causal effect of each CpG to a lipid measure.

**Reporting summary**. Further information on research design is available in the Nature Research Reporting Summary linked to this article.

## Data availability
The methylation QTL association results as well as the full summary statistics for the meta-analysis of the epigenome wide association study performed within each and across all racial/ethnic groups combined for all four models and all three lipid traits are available at [https://doi.org/10.5061/dryad.qfttdz0d8]. All other relevant data supporting the key findings of this study are available within the article and its Supplementary Information files or from the corresponding authors upon reasonable request. A reporting summary for this Article is available as a Supplementary Information file.

## Code availability
The following freely available software was used for analyses: R (V.3.4.4 and later, https://www.r-project.org/) including minfi, wateRmelon, and qqman packages; Python 2.7.5 (https://www.python.org/downloads/); METASOFT (V.2.0.1, http://genetics.cs.ucla.edu/meta_jemdoc/).

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

## Acknowledgements

We thank our Amish community and research volunteers for their long-standing partnership in research, and acknowledge the dedication of our Amish liaisons, field workers and the Amish Research Clinic staff, without which these studies would not have been possible. We thank the staff and participants of the ARIC study for their important contributions. We would also like to thank the families that participated in the GENOA study. The authors are grateful to the Rotterdam Study participants, the staff involved with the Rotterdam Study and the participating general practitioners and pharmacists. The generation and management of the Illumina 450 K methylation array data (EWAS data) for the Rotterdam Study was executed by the Human Genotyping Facility of the Genetic Laboratory of the Department of Internal Medicine, Erasmus MC, the Netherlands. We thank Mr. Michael Verbiest, Ms. Mila Jhamai, Ms. Sarah Higgins, Mr. Marijn Verkerk, and Lisette Stolk PhD for their help in creating the methylation database. We thank Ms. Mila Jhamai, Ms. Sarah Higgins, Marjolein Peters, MSc, Mr. Marijn Verkerk and Jeroen van Rooij, MSc for their help in creating the RNA array expression database. Infrastructure for the CHARGE Consortium is supported in part by the National Heart, Lung, and Blood Institute grant R01HL105756. The Atherosclerosis Risk in Communities study has been funded in whole or in part with Federal funds from the National Heart, Lung, and Blood Institute, National Institutes of Health, Department of Health and Human Services (contract numbers HHSN268201700001I, HHSN268201700002I, HHSN268201700003I, HHSN268201700004I, and HHSN268201700005I). Funding was also supported by 5RC2HL102419 and R01NS087541. The CHS research was supported by NHLBI contracts HHSN268201200036C, HHSN268200800007C, N01HC55222, N01HC85079, N01HC85080, N01HC85081, N01HC85082, N01HC85083, N01HC85086, 75N92021D00006; and NHLBI grants U01HL080295, R01HL087652, R01HL092111, R01HL105756, R01HL103612, R01HL111089, R01HL116747, R01HL120393, and U01HL130114 with additional contribution from the National Institute of Neurological Disorders and Stroke (NINDS). Additional support was provided through R01AG023629 from the National Institute on Aging (NIA) as well as Laughlin Family, Alpha Phi Foundation, and Locke Charitable Foundation. A full list of principal CHS investigators and institutions can be found at CHS-NHLBI.org. The provision of genotyping data was supported in part by the National Center for Advancing Translational Sciences, CTSI grant UL1TR000124, and the National Institute of Diabetes and Digestive and Kidney Disease Diabetes Research Center (DRC) grant DK063491 to the Southern California Diabetes Endocrinology Research Center. The content is solely the responsibility of the authors and does not necessarily represent the official views of the National Institutes of Health. The Framingham Heart Study is funded by National Institutes of Health contract N01- HC-25195. The laboratory work for this investigation was funded by the Division of Intramural Research, National Heart, Lung, and Blood Institute, National Institutes of Health and an NIH Director's Challenge Award (D. Levy, Principal Investigator). The GOLDN epigenetics study is funded by the NIH National Heart, Lung, and Blood Institute grant R01 HL104135-01. Support for the Genetic Epidemiology Network of Arteriopathy (GENOA) was provided by the National Heart, Lung and Blood Institute (HL054457, HL100185, HL119848, and HL133221) of the National Institutes of Health. German Research Center for Environmental Health, which is funded by the German Federal Ministry of Education and Research (BMBF) and by the State of Bavaria. Furthermore, KORA research was supported within the Munich Center of Health Sciences (MC-Health), Ludwig-Maximilians-Universität, as part of LMUinnovativ. This work was supported by a grant (WA 4081/1-1) from the German Research Foundation. The TwinsUK epigenetic study received support from the ESRC (ES/N000404/1). The TwinsUK study funded by the Wellcome Trust; European Community's Seventh Framework Programme (FP7/2007–2013); National Institute for Health Research (NIHR)- funded BioResource, Clinical Research Facility and Biomedical Research Centre based at Guy's and St Thomas' NHS Foundation Trust in partnership with King's College London. SNP genotyping was performed by The Wellcome Trust Sanger Institute and National Eye Institute via NIH/CIDR. The PIVUS study was supported by Swedish Research Council (Grant no. 2012-1397), Knut och Alice Wallenberg

Foundation (Grant no. 2013.0126), Swedish Heart-Lung Foundation (grant no. 20140422), and Swedish Diabetes Foundation (Grant no. 2013-024). The Rotterdam Study EWAS data was funded by the Genetic Laboratory of the Department of Internal Medicine, Erasmus MC, and by the Netherlands Organization for Scientific Research (NWO; project number 184021007) and made available as a Rainbow Project (RP3; BIOS) of the Biobanking and Biomolecular Research Infrastructure Netherlands (BBMRI-NL). The generation and management of RNA-expression array data for the Rotterdam Study was executed and funded by the Human Genotyping Facility of the Genetic Laboratory of the Department of Internal Medicine, Erasmus MC, the Netherlands. The Women's Health Initiative data, WHI-BA23 and WHI-EMPC, were generated through an NHLBI Broad Agency Announcement contract (HHSN268201300006C) and a National Institute of Environmental Health Sciences grant (R01-ES020836). The WHI program is funded by the National Heart, Lung, and Blood Institute, National Institutes of Health, U.S. Department of Health and Human Services through contracts HHSN268201100046C, HHSN268201100001C, HHSN268201100002C, HHSN268201100003C, HHSN268201100004C, and HHSN271201100004C.

## Author contributions

M.A.J., M.M., M.R.I., and T.L.A. contributed to study design. M.A.J., R.W., R.G., R.J., E.S., X.Z., K.V.E.B. AND, A.K.H., T.Z., E.C.M., J.S., T.M.B., J.A.B., M.E.M., J.R.O., C.Y., P.S., W.Z., E.B.W., S.G. contributed to cohort-specific data analyses. M.A.J. contributed to meta-analyses of EWAS and mQTL, and Mendelian randomization analyses. M.A.J. and T.L.A. contributed to interpretation of the results and writing of manuscript. M.M., R.W., R.G., K.V.E.B., T.M.B., T.M., B.M.P., E.I., M.F., J.T.B., L.L., M.W. contributed to critical review of manuscript. E.B., M.G., W.G., L.P., M.M.N., T.M., C.G., A.P., E.B.W., J.A.S., K.D., J.v.M., A.U., M.A.I., M.G., D.Z., L.L., S.L., T.D.S., Y.I.C., C.D., A.R.S., D.M.A., S.H., P.S.T., S.K., B.M.P., N.S., J.T.B., E.I., W.C., A.D., D.A., M.W., L.H., E.A.W., A.B., D.L., M.F., M.R.I., and T.L.A. contributed to cohort design and management, and data collection.

## Competing interests

Alan R Shuldiner is an employee of Regeneron Pharmaceuticals, Inc. Bruce M Psaty serves on the DSMB of a clinical trial funded by the manufacturer (Zoll LifeCor) and on the Steering Committee of the Yale Open Data Access Project funded by Johnson & Johnson. Kim Valeska Emilie Braun works in ErasmusAGE, a center for aging research across the life course funded by Nestlé Nutrition (Nestec Ltd.), Metagenics Inc. and AXA.

All other authors declare no competing interests. The funders had no role in design and conduct of the study; collection, management, analysis, and interpretation of the data; and preparation, review or approval of the manuscript.

## Additional information

[1]Department of Epidemiology, University of Michigan School of Public Health, Ann Arbor, MI, USA. [2]Department of Medicine, Stanford University School of Medicine, Stanford, CA, USA. [3]Population Sciences Branch, National Heart, Lung, and Blood Institute, National Institutes of Health, Bethesda, MD, USA. [4]Department of Cardiology, Boston Children's Hospital, Boston, MA, USA. [5]Research Unit Molecular Epidemiology, Institute of Epidemiology, Helmholtz Zentrum München, German Research Center for Environmental Health, Munich, Germany. [6]Institute of Epidemiology, Helmholtz Zentrum München, German Research Center for Environmental Health, Munich, Germany. [7]Department of Epidemiology, University of North Carolina, Chapel Hill, NC, USA. [8]Hebrew SeniorLife, Beth Israel Deaconess Medical Center, Harvard Medical School, Boston, MA, USA. [9]The Brown Foundation Institute of Molecular Medicine, McGovern Medical School, The University of Texas Health Science Center at Houston, Houston, TX, USA. [10]Department of Epidemiology, Erasmus MC University Medical Center, Rotterdam, The Netherlands. [11]Department of Epidemiology, School of Public Health, University of Alabama at Birmingham, Birmingham, AL, USA. [12]Department of Medical Sciences, Molecular Epidemiology and Science for Life Laboratory, Uppsala University, Uppsala, Sweden. [13]Department of Epidemiology, Tulane University School of Public Health and Tropical Medicine, New Orleans, LA, USA. [14]Department of Twin Research and Genetic Epidemiology, School of Life Course Sciences, King's College London, London, UK. [15]GENYO, Center for Genomics and Oncological Research Pfizer/University of Granada/Andalusian Regional Government, Granada, Spain. [16]Department of Population Health Sciences, University of Utah, Salt Lake City, UT, USA. [17]Cardiovascular Health Research Unit, Departments of Medicine and Biostatistics, University of Washington, Seattle, WA, USA. [18]Cardiovascular Health Research Unit, Department of Medicine, University of Washington, Seattle, WA, USA. [19]Division of Endocrinology, Diabetes, and Nutrition, University of Maryland School of Medicine, Baltimore, MD, USA. [20]Program for Personalized and Genomic Medicine, University of Maryland School of Medicine, Baltimore, MD, USA. [21]School of Public Health, University of Texas Health Science Center at Houston, Huston, TX, USA. [22]Division of Biostatistics, School of Public Health, University of Minnesota, Minneapolis, MN, USA. [23]Institute of Genetic Epidemiology, Helmholtz Zentrum München, German Research Center for Environmental Health, Neuherberg, Germany. [24]IBE, Faculty of Medicine, LMU Munich, Munich, Germany. [25]Institute of Medical Biostatistics, Epidemiology and Informatics (IMBEI), University Medical Center, Johannes Gutenberg University, Mainz, Germany. [26]Institute of Human Genetics, Helmholtz Zentrum München, German Research Center for Environmental Health, Munich, Germany. [27]Institute of Human Genetics, Technical University Munich, Munich, Germany. [28]Survey Research Center, Institute for Social Research, Ann Arbor, MI, USA. [29]Department of Internal Medicine, Rush University Medical Center, Chicago, IL, USA. [30]Department of Internal Medicine, Erasmus MC University Medical Center, Rotterdam, The Netherlands. [31]School of Biomedical Informatics, University of Texas Health Science Center at Houston, Houston, TX, USA. [32]Department of Epidemiology and Biostatistics, School of Public Health, Peking University Health Science Center, Beijing, China. [33]Institute for Translational Genomics and Population Sciences, Los Angeles Biomedical Research Institute, and Department of Pediatrics, Harbor-UCLA Medical Center, Torrance, CA, USA. [34]HudsonAlpha Institute for Biotechnology, Huntsville, AL, USA. [35]Department of Human Genetics, David Geffen School of Medicine, University of California Los Angeles, Los Angeles, CA, USA. [36]Department of Biostatistics, Fielding School of Public Health, University of California Los Angeles, Los Angeles, CA, USA. [37]VA Palo Alto Healthcare System, Palo

Alto, CA, USA. [38]Stanford Cardiovascular Institute, Stanford University, Stanford, CA, USA. [39]Cardiovascular Health Research Unit, Departments of Epidemiology, Medicine, and Health Services, University of Washington, Seattle, WA, USA. [40]Kaiser Permanente Washington Health Research Institute, Seattle, WA, USA. [41]Cardiovascular Health Research Unit, Division of Cardiology, University of Washington, Seattle, WA, USA. [42]Stanford Diabetes Research Center, Stanford University, Stanford, CA, USA. [43]Department of Biostatistics and Epidemiology, MRC Centre for Environment and Health, School of Public Health, Imperial College, London, UK. [44]Department of Preventive Medicine, Feinberg School of Medicine, Northwestern University, Chicago, IL, USA. [45]Department of Medicine, University of North Carolina, Chapel Hill, NC, USA. [46]Department of Environmental Health Sciences, Harvard T.H. Chan School of Public Health, Boston, MA, USA. [47]Department of Environmental Health Sciences, Columbia University Mailman School of Public Health, New York, NY, USA. [48]Framingham Heart Study, Framingham, MA, USA. ✉email: mina.jhun82@gmail.com; tassimes@stanford.edu

