## [Peer Review File · Nature Communications]

Reviewers' comments:

Reviewer #1 (Remarks to the Author):

Please see attachment

Reviewer #2 (Remarks to the Author):

The authors performed a large multi-ethnic epigenome-wide association study meta-analysis of leukocyte DNA methylation and blood lipids (HDL, LDL, and triglycerides). This is an important piece of work because I believe it is the largest EWAS of lipids to date, it compares and combines EWAS results from multiple race/ethnic groups, and it identifies a number of novel methylation sites associated with blood lipids. I mainly have suggestions to improve the clarity of the paper. In particular, the analytic design (4 models, 3 populations + trans-ethnic meta-analysis, functional follow-up analysis for a subset of results) could be presented more clearly. I would recommend to make genome-wide summary statistics of the meta-analysis available.

Specific comments:

Abstract

- The description of the comparison across race/ethnic groups could be improved by providing information on the correlation of effect sizes or consistency of direction of effects. Currently the abstract only mentions how many CpGs are significant in multiple ethnic groups, but this comparison is influenced by differences in samples size (thus power) between groups.
- The term 'corresponding gene' sounds odd to me, given that a sizeable proportion of probes from the Illumina 450k array are located intergenic. This raises the question: what is 'the corresponding gene' for intergenic CpGs, or did all CpGs happen to be located within a gene?
- 'that are independent of loci identified through GWAS'. The phrasing 'independent' is somewhat ambiguous, and it appears from the results section that there is some overlap between loci found in the EWAS and GWAS of lipids.

Results

- It is common to report estimates of inflation (λ). Could this information be added?
- The way in which the results from the different models and populations are presented lacks a clear structure. Firstly, it is unclear, without reading the methods, what the 4 models are. It seems that the results section only mentions results from 3 models. Furthermore, it is not clear that model 1 and 2 correct for lipid medication.
- Strangely, it is not mentioned at all in the results and discussion section that model 2 not only includes BMI as an additional covariate, but also smoking. This should be clarified in the results and discussion.
- It is unclear why the results section only describes model 1 and 2 results for Europeans, but not for African Americans and Hispanics.
- It is not clear on which model the trans-ethnic meta-analyses were performed.
- "Through trans-ethnic meta analyses of the same model, we additionally identified 49, 24, and 119 significance ($P < 1.09 \times 10^{-7}$) CpG-lipid level associations for HDL, LDL, and TG,

respectively, of which 46, 22, and 118 were novel”.

> Here, the term ‘novel’ is confusing to me. Do the authors mean novel compared to their race/ethnic group-specific analyses or novel compared to previous literature? This also raises the questions how many of the loci detected in the race/ethnic group-specific analyses have not been reported before in previous EWA studies of lipids.

- In the section ‘comparison across race/ethnic groups’ it is unclear which of the 4 models is described in this section.

- It is unclear how the overlap with prior GWA studies was done exactly; in particular why and how the overlap was examined at the gene level. Are intergenic CpGs and intergenic SNPs excluded from this comparison?

- It would be informative to mention on which populations (race/ethnicity) previous EWAS and GWAS were performed.

- ‘A total of 6 genes had both CpGs and SNPs significantly associated with a blood lipid’. Instead of only describing the overlap at the gene level, it would be informative to describe the number of CpGs from the EWAS that map to GWAS loci.

- ‘Many of the CpGs identified in our lipid EWAS have been previously identified in EWAS of BMI’. >Please report the exact number of CpGs (or %).

- ‘Among EA, 34 out of 106 genes have been identified in EWAS of other phenotypes’. > Why is this only described for Europeans?

- The authors only discuss BMI as an explanation for attenuation of effect sizes in model 2 and 4, and only describe the overlap with CpG sites from previous BMI EWA studies, while model 2 also corrects for smoking. What is the overlap with smoking-associated CpG sites and to what extent is the association between lipids and methylation possibly driven by smoking?

- It is unclear why the authors chose to use a random effects meta-analysis for the EWAS results, and a fixed effects meta-analysis for the mQTL results.

- Note that the term mQTL refers to the SNP, not the CpG. In the manuscript, mQTL is currently used to refer to CpGs at least once.

- “Out of the 190 significant mQTLs common to both our study and the mQTL DB, 51 (27%) were found to be mQTLs in datasets spanning across the life course in the mQTL DB including birth, childhood, adolescence, pregnancy, and middle age”. > It is unclear to me if 190 refers to SNPs, CpGs, or SNP-CpG pairs.

- The sample size (total number of participants) of the mQTL analysis, expression analysis and Mendelian Randomization Analysis should be reported in the results section.

- The results section does not mention on which populations (EA, AA or HISP) the expression and Mendelian Randomization analyses are performed.

- It would be informative to report the variance in lipid levels explained by the GRSs.

- The overlap of SNPs that are included in the GRS for the different lipids, and mQTLs should be discussed. Related to this point, pleiotropy should be discussed as a possible limitation of MR analyses.

- ‘We implemented inverse-weighted MR method and MR-egger when >2 mQTLs were available for a given CpG.’ > For how many CpGs was this the case? (this should be clarified in the text)

- It is unclear to me why the 2 directions of causation were tested using different methods (lipids to methylation uses a GRS approach, without consideration of pleiotropy it seems, while methylation to lipids is tested using MR-egger, which is based on summary statistics and does take pleiotropy into account).

- The authors might want to consider adding Manhattan plots from the other groups to figure 1 (it currently only shows results from the analyses in Europeans), because the trans-ethnic component is a major asset of this study.

- In figure 2, it is unclear to me why HISP and AA are only compared to EA, and not to each

other (AA versus HISP).

- The Venn diagrams are informative (figure 3). Would it be useful to include similar Venn diagrams to illustrate the overlap of findings across: European, African, Hispanic, and the trans-ethnic meta-analysis? Perhaps in that case it is more practical to show numbers of CpGs instead of gene names.
- Which standard deviation is shown in table 2?
- Supplemental tables could be organized better. Some have multiple tabs or multiple tables on one tab, and there are no legends. For example, in supplemental table 2, it is unclear what exactly is the purpose of the two sub-tables on the first tab and in table S3, what does TE.fixed and num_studies mean?

Discussion

- How do the authors interpret the weak correlation of effect sizes across ethnicities for LDL compared to the relatively strong correlation of effect sizes across ethnicities for HDL and triglycerides?
- I think that 'concordance of effects' should be 'concordance of the direction of effects'.
- There seems to be a discrepancy between the number of CpGs that are associated with gene expression levels in the results section (7) and the discussion section (5).
- "Third, we found the methylation status of CpGs in CPT1A, a gene that initiates the oxidation of long-chain fatty acids, to be influenced by blood levels of TG through our MR analysis consistent with findings from a previous EWAS of this locus with blood pressure"
> This sentence is unclear to me. Did the EWAS of blood pressure also find a causal effect of TG on methylation at this locus? Or a causal effect of blood pressure?
- How do the results of the MR analyses compare to findings from the previous MR study on lipids and methylation by Dekkers et al (reference 14)?
- Is it possible to add a reference for the sentence "While circulating leucocytes are likely to exert at least partial direct control over blood lipid levels"?

Methods

- The description of correction for family structure is very brief. I expected to find more details about this in the supplemental methods (i.e. what type of family relationships are present in which cohort, and if multiple degrees of relatedness are present, are these modeled by inclusion of multiple random effects?)
- Are the same white blood cell proportions included in all analyses (i.e. EWAS, mQTL, eQTL, MR?). From the description of the EWAS analyses, it seems that all cohorts used Houseman estimates, while from the description of the mQTL analyses, it seems that some cohorts used measured cell counts while other used Houseman estimates.
- In formula for the GRS, it seems odd to me that the allele dosages were not multiplied by the weights for each SNP from the GWAS. Is this an error?

Signed

Jenny van Dongen

Reviewer #1 Attachment

Jhun *et al.* have performed a multi-ethnic EWAS of peripheral blood leukocytes and blood lipids. Dyslipidaemia is clearly a significant cardiovascular disease risk factor. Pioneering papers have already explored Lipids in peripheral blood DNA methylation such as Irvin *et al.*¹ and Dekkers *et al.*², including many of the other cohorts studies included here and others as well. This meta-analysis has involved 15 EWAS studies of ~16k individuals that includes 3 ethnic groups with 12 cohorts European, 7 cohorts African American, 2 cohorts Hispanic. All the studies are peripheral blood derived DNA except one cohort (GOLDN) CD4⁺ T cells only. The largest European grouping has the most power and identified the most 'novel' findings. Added the other population groups also increases sample size, but with the caveat of increasing the genetic heterogeneity and potentially confounding. The difficulty here is ascertaining how 'novel' these results are when the phenotype is so inter-related with other metabolic syndrome traits, so many of these changes are seen in other trait EWAS, such as BMI etc. Nevertheless, increasing the cohort sizes clearly leads to some more significant findings reaching genome-wide significance levels. Points for the authors to consider are below:

Major

1. It would be of interest to assess how much these meta-analyses are actually the appropriate way ahead for EWAS analysis with just increased sample size – particularly including different populations. Whether in fact there would be more power in investigating smaller and more homogenous populations, but with those with more extreme ends of a phenotype? Could the authors explore this in the European-only group by a case-control analysis of those with very high and the lowest lipid levels. Although, they have not recruited for extreme dyslipidaemia, so may not be able to test this approach properly by not having enough extreme phenotypes. However, could the authors estimate and comment on the power of this alternative and potentially cheaper approach.
2. Could to authors explain further why there was not a high correlation between LDL effect sizes? (pg 160)
3. Overlap analysis - Should discuss individual probe overlap not just the 34 out of 104 gene overlaps (line 186) – as this can be greatly influenced by gene size and probe density variations per gene.
4. The eQTM analysis – could the authors include whether all these results were novel findings - and was there any evidence of these CpGs residing in any critical transcription factor binding sites? Was there any evidence of these CpGs being within broader DMRs within these regions?
5. For the Mendelian Randomisation results – what was the biological mechanism proposed for the second analysis of DNA methylation on blood lipid levels effect?

Minor

1. Introduction, line 123 - *Cis*-methylation quantitative trait loci abbreviation *cis*-mQTL not *cis*-QTL?
 2. Line 202 and elsewhere: Expression quantitative trait methylations eQTM – should this be ‘methylation’ not ‘methylations’?
-
1. Irvin, M.R. *et al.* Epigenome-Wide Association Study of Fasting Blood Lipids in the Genetics of Lipid-Lowering Drugs and Diet Network Study. ***Circulation*** 130, 565-572 (2014).
 2. Dekkers, K.F. *et al.* Blood lipids influence DNA methylation in circulating cells. ***Genome Biol*** 17, 138 (2016).

Reviewers' comments:

Reviewer #1 (Remarks to the Author):

Summary: Jhun et al. have performed a multi-ethnic EWAS of peripheral blood leukocytes and blood lipids. Dyslipidemia is clearly a significant cardiovascular disease risk factor. Pioneering papers have already explored Lipids in peripheral blood DNA methylation such as Irvin et al., and Dekkers et al., including many of the other cohorts studies included here and others as well. This meta-analysis has involved 15 EWAS studies of ~16k individuals that includes 3 ethnic groups with 12 cohorts European, 7 cohorts African American, 2 cohorts Hispanic. All the studies are peripheral blood derived DNA except one cohort (GOLDN) CD4+ T cells only. The largest European grouping has the most power and identified the most 'novel' findings. Added the other population groups also increases sample size, but with the caveat of increasing the genetic heterogeneity and potentially confounding. The difficulty here is ascertaining how 'novel' these results are when the phenotype is so inter-related with other metabolic syndrome traits, so many of these changes are seen in other trait EWAS, such as BMI etc. Nevertheless, increasing the cohort sizes clearly leads to some more significant findings reaching genome-wide significance levels. Points for the authors to consider are below:

Major:

1. It would be of interest to assess how much these meta-analyses are actually the appropriate way ahead for EWAS analysis with just increased sample size – particularly including different populations. Whether in fact there would be more power in investigating smaller and more homogenous populations, but with those with more extreme ends of a phenotype? Could the authors explore this in the European-only group by a case-control analysis of those with very high and the lowest lipid levels. Although, they have not recruited for extreme dyslipidemia, so may not be able to test this approach properly by not having enough extreme phenotypes. However, could the authors estimate and comment on the power of this alternative and potentially cheaper approach.

We appreciate the reviewer's suggestions of an alternative and potentially cheaper approach of profiling a subset of samples with extreme lipid levels. As understood, participants in this study were not recruited nor profiled based on extreme dyslipidemia. Depending on the cohort, participants with high cholesterol were treated to a varying degree with lipid lowering medications truncating the tails of the distribution even more. Furthermore, extreme values (defined by >5 standard deviations from the mean of a blood lipid) in each cohort were excluded in case these entries represented data errors (could be due to an error in measurement or entering the data value in computing system).

In general, a continuous variable is known to have a higher statistical power compared to a dichotomized variable [see Altman DG, Royston P. The cost of dichotomising continuous variables. *BMJ*. 2006;332(7549):1080. doi:10.1136/bmj.332.7549.1080]. Focusing on extreme values often reduce the total sample size resulting in even more reduced statistical power.

We agree with the reviewer's opinion that it could be more economically efficient to measure DNA methylation for a smaller number of subjects with extreme values when resources are limited for DNA methylation measures and enough extreme phenotypes have already been collected. If the extreme values have not been collected, the efficiency will depend on how much resources need to be spent identifying subjects with extreme values. For this study, all the participated cohorts already had DNA methylation measured for other purposes hence we tried to maximize the statistical power by utilizing all the samples rather than using subset of samples with extreme lipid levels.

2. Could to authors explain further why there was not a high correlation between LDL effect sizes? (pg 160)

Thank you for this comment which pushed us to better understand this discrepancy. As shown in Figure 2, there were substantially smaller number of CpGs identified for LDL compared to HDL or Triglycerides (TG). To improve our insight regarding the correlation of betas among the three race/ethnic groups for LDL, we investigated 122 CpGs ($P < 1.09 \times 10^{-5}$ in any of the race/ethnic groups) instead of 21 CpGs ($P < 1.09 \times 10^{-7}$, the original Figure 2b). The Pearson correlation of betas between EA and AA increased from -0.20 to 0.49 and the correlation between EA and HA increased from 0.20 to 0.21.

In addition, only HDL and TG were natural log-transformed due to the skewed distributions. Since natural log transformation is nonlinear, it could impact the correlation of the betas. To examine this, we also natural log-transformed betas of LDL for the 122 CpGs ($P < 1.09 \times 10^{-5}$) and calculated Pearson correlations. The Pearson correlation of betas between EA and AA increased from -0.20 to 0.37 and the correlation between EA and HA increased from 0.20 to 0.39.

In the revised manuscript, we added the beta comparison plots for 122 CpGs ($P < 1.09 \times 10^{-5}$, with and without natural log transformation) as **Supplementary Figure 4** and the following sentences in the "Comparison of results across race/ethnic groups" section of Results: "We calculated the correlations coefficients and estimated regression slopes for a higher number of CpGs (122 CpGs with $P < 1.09 \times 10^{-5}$) before and after natural log transformation to further explore whether the differences in correlations and regression slopes between LDL and HDL/TG could be a consequence of having much smaller number of significant CpGs and/or the use of a log transformed lipid measure. For these analyses, we found both higher correlation of betas (0.21 to 0.49) and regression slopes (0.34 to 0.47) for LDL although still not as high as those observed for HDL and TG (**Supplementary Figure 4**)."

3. Overlap analysis – Should discuss individual probe overlap not just the 34 out of 104 gene overlaps (line 186) – as this can be greatly influenced by gene size and probe density variations per gene.

Thank you for the comment. In the original analysis, we also compared the individual CpG probes. In the revised manuscript we added relevant information for individual probe overlap by replacing the original sentence "Among EA, 34 out of 106 genes have been identified in EWAS of other phenotypes." with "Among EA, 37 out of 106 genes (39 out of 164 CpGs) have been identified in EWAS of other phenotypes." (In the revised version, we added one more comparison with C-reactive protein EWAS resulting in 37 overlapping genes.)

4. The eQTM analysis – could the authors include whether all these results were novel findings – and was there any evidence of these CpGs residing in any critical transcription factor binding sites? Was there any evidence of these CpGs being within broader DMRs within these regions?

Yes, the CpGs identified in the eQTM analysis were annotated for location in a gene, CpG island, enhancer, and DNase I hypersensitive site (DHS) in Table 2. In the eQTM result, cg06500161 is residing in Enhancer region, cg07397296 is located in DHS region, and cg14476101 is located in reprogramming-specific differentially methylated region (RDMR). In the revised version of the manuscript, we added the following annotations in the eQTM results (**Supplementary Table 5**): Have been identified in previous lipid EWAS, DMR, Enhancer, and DHS.

5. For the Mendelian Randomization results – what was the biological mechanism proposed for the second analysis of DNA methylation on blood lipid levels effect?

We thought smoking, diet, exercise, and/or other environmental factors could have an influence on DNA methylation levels resulting in changed in blood lipid levels.

Minor

1. Introduction, line 123 – Cis-methylation quantitative trait loci abbreviation cis-mQTL not cis-QTL?

Thanks for finding the typo. It is corrected in the revised version of the manuscript.

2. Line 202 and elsewhere: Expression quantitative trait methylations eQTM – should this be ‘methylation’ not ‘methylations’?

Thanks for the comment. We updated ‘methylations’ to ‘methylation’ in the revised version of the manuscript.

Reviewer #2 (Remarks to the Author):

The authors performed a large multi-ethnic epigenome-wide association study meta-analysis of leukocyte DNA methylation and blood lipids (HDL, LDL, and triglycerides). This is an important piece of work because I believe it is the largest EWAS of lipids to date, it compares and combines EWAS results from multiple race/ethnic groups, and it identifies a number of novel methylation sites associated with blood lipids. I mainly have suggestions to improve the clarity of the paper. In particular, the analytic design (4 models, 3 populations + trans-ethnic meta-analysis, functional follow-up analysis for a subset of results) could be presented more clearly. I would recommend to make genome-wide summary statistics of the meta-analysis available.

Thank you for this suggestion. We do plan on making the summary statistics of all CpG associations available to other investigators (i.e. genome wide) through a publicly accessible portal (e.g. dbGAP/CHARGE). We plan on making these results available for each of the 3 race/ethnic group analyses as well as the trans-ethnic analysis.

Specific comments:

Abstract

- The description of the comparison across race/ethnic groups could be improved by providing information on the correlation of effect sizes or consistency of direction of effects. Currently the abstract only mentions how many CpGs are significant in multiple ethnic groups, but this comparison is influenced by differences in samples size (thus power) between groups.

Thanks for the comment. In the revised manuscript, we have added the following sentence to the abstract “For all three lipid fractions, we found high concordance in the direction of effects observed in Europeans versus African Americans and, separately, Europeans versus Hispanics. Analogous comparisons of the correlations of effects sizes were high for TG and HDL and modest for LDL.”

- The term ‘corresponding gene’ sounds odd to me, given that a sizeable proportion of probes from the Illumina 450k array are located intergenic. This raises the question: what is ‘the corresponding gene’ for intergenic CpGs, or did all CpGs happen to be located within a gene?

In the following phrase of the manuscript, “seven CpGs showing association with the expression of the corresponding gene and CpGs in *CPT1A*“, the corresponding gene means the gene annotated by Illumina to the CpG (in this case *CPT1A*). CpGs were annotated to genes by Illumina using the following rules: those located within 1500bp upstream of transcription start site (TSS1500), TSS200, 5’UTR, 1st exon, gene body, or 3’UTR of a gene were annotated to that gene. All other intergenic CpGs were not annotated to a gene. Out of the 30 CpGs in Table2, five without a gene name were located in an intergenic region. In the revised manuscript, we included the rules on how Illumina annotated CpGs to genes and changed the wording in the abstract and the rest of the manuscript from “corresponding” to “annotated”.

- ‘that are independent of loci identified through GWAS’. The phrasing ‘independent’ is somewhat ambiguous, and it appears from the results section that there is some overlap between loci found in the EWAS and GWAS of lipids.

In the abstract of the revised manuscript, we rephrase the sentence to “and the location of loci were largely non-overlapping with loci identified through GWAS”.

Results

- It is common to report estimates of inflation (lambda). Could this information be added?

We acknowledge that it is important to report estimates of inflation (lambda). We observed inflated lambda in some of our EWAS. To avoid any false positive findings due to the inflation on the statistic, we implemented genomic control during the meta-analysis (described in the “Meta-analysis section of methods). As a result of the genomic control, there was no inflation of lambda after meta-analysis with all lambdas being 1. In the revised manuscript, we added the lambda (before and after genomic control) of all the four models in stratified and all race/ethnic group meta-analyses as **Supplementary Table 8**.

- The way in which the results from the different models and populations are presented lacks a clear structure. Firstly, it is unclear, without reading the methods, what the 4 models are. It seems that the results section only mentions results from 3 models. Furthermore, it is not clear that model 1 and 2 correct for lipid medication.

We describe the four models using equations that includes all the covariates in the Methods section. To avoid any confusion, we also added the covariates information of the models in Results. In the revised manuscript, the following four modifications were made in the second paragraph of “Epigenome-wide association study (EWAS) stratified by race/ethnic group” section of Results:

- (1) “(model1)” -> “(model1 adjusted for age, sex, smoking, lipid medication, four SNP PCs, estimated cell proportions, plate, row, and column of plate) (Methods)”
- (2) “(model2)” -> “(model2 additionally adjusted for BMI)”
- (3) “Excluding participants taking any lipid lowering medication decreased the sample size by 18% and decreased power but the effect estimates remained similar (Methods)” -> “Excluding participants taking any lipid lowering medication decreased the sample size by 18% and decreased power but the effect estimates remained similar (models 3 and 4 adjusted for the same set of covariates of models 1 and 2, respectively, with the exception of adjustment for the use of lipid medications)”
- (4) “Among EA, we identified 74, 15, and 86 CpGs significantly associated ($P < 1.09 \times 10^{-7}$) with HDL, LDL, and TG, respectively, using this most conservative model that excluded statin users and adjusted for BMI (Methods)” -> “Among EA, we identified 74, 15, and 86 CpGs significantly associated ($P < 1.09 \times 10^{-7}$) with HDL, LDL, and TG, respectively, using this most conservative model that excluded statin users and adjusted for BMI (model 4, Methods)”

- Strangely, it is not mentioned at all in the results and discussion section that model 2 not only includes BMI as an additional covariate, but also smoking. This should be clarified in the results and discussion.

We apologize for not making it initially clear that smoking status was used as a covariate for all the models 1-4 as described in “Epigenome-wide association study (EWAS)” section of Methods.

“**Model 1:** Methylation (beta) ~ lipid + age + sex + smoking + lipid medication + SNP PC1-4 + WBC (or estimates from Houseman’s method) + (1|plate) + (1|row) + (1|column)”

Model 2: Methylation (beta) ~ lipid + age + sex + BMI + smoking + lipid medication + SNP PC1-4 + WBC (or estimates from Houseman's method) + (1|plate) + (1|row) + (1|column)"

To avoid a confusion, we added description of the models in "Epigenome-wide association study (EWAS) stratified by race/ethnic group" section of Results

- (1) "(model1)" -> "(model1 adjusted for age, sex, smoking, lipid medication, four SNP PCs, estimated cell proportions, plate, row, and column of plate) (Methods)"
- (2) "(model2)" -> "(model2 additionally adjusted for BMI)"

- It is unclear why the results section only describes model 1 and 2 results for Europeans, but not for African Americans and Hispanics.

In the revised version of the manuscript, we added the following descriptions of AA and HA to "Epigenome-wide association study (EWAS) stratified by race/ethnic group" section of Results to address this omission: "Among AA, we identified 34, 7, and 76 CpGs in model 1 for HDL, LDL, and TG, respectively, and the numbers decreased to 9, 7, and 55 with a further adjustment with BMI (model2). For HISP, we identified 2, 0, and 6 CpGs in model 1 for HDL, LDL, and TG, respectively, and the number decreased to 0 for HDL."

- It is not clear on which model the trans-ethnic meta-analyses were performed.

We performed the trans-ethnic meta-analyses on all 4 models. The number of significant CpGs is summarized in "Number of significant CpGs" sheet of Supplementary Table2. To make it clear for readers, we add a sentence in "Meta-analysis" subsection of the Methods section: "These meta-analyses were performed for each of the 4 models, respectively."

- "Through trans-ethnic meta analyses of the same model, we additionally identified 49, 24, and 119 significance ($P < 1.09 \times 10^{-7}$) CpG-lipid level associations for HDL, LDL, and TG, respectively, of which 46, 22, and 118 were novel".
> Here, the term 'novel' is confusing to me. Do the authors mean novel compared to their race/ethnic group-specific analyses or novel compared to previous literature? This also raises the questions how many of the loci detected in the race/ethnic group-specific analyses have not been reported before in previous EWA studies of lipids.

We report "novel" findings in our trans-ethnic analyses 1) that are not the novel findings in our race/ethnic group-specific analyses and 2) that are not reported in the previous literature (see "Number of novel associations" sheet in Supplementary Table 2). This is the reason why we used the term "additional" in the following sentence in our abstract: "an additional 186 (46+22+118) novel associations were identified through trans-ethnic meta-analysis". To avoid a confusion, we revised the sentence in our results section as follows: "of which 46, 22, and 118 were novel when compared to both our race/ethnic specific analyses and the literature."

- In the section 'comparison across race/ethnic groups' it is unclear which of the 4 models is described in this section.

This section describes the results of the main model 4. To make it clear, we added “(model4)” to the second sentence of the “Comparison of results across race/ethnic groups” section of Results in the revised manuscript.

- It is unclear how the overlap with prior GWA studies was done exactly; in particular why and how the overlap was examined at the gene level. Are intergenic CpGs and intergenic SNPs excluded from this comparison?

Most DNA methylation occurs at Cytosine followed by Guanine while SNPs can happen at most of locations. For the Illumina 450K methylation array used in our study, the majority of CpGs do not overlap with SNPs by design. This is because the DNA methylation signal could be influenced by a SNP if there is a SNP at (C->A,T,G mutation resulting in no Cytosine for methylation) or near (DNA methylation probe affinity decreases due to a SNP on the probe) the CpG site. This is why we examined the overlap between EWAS and GWAS at the gene level.

Even though a SNP and a CpG are not at the exact same location, a SNP and a CpG could have an effect on the transcription of the same gene. For instance, we can assume a missense SNP in the first exon and a CpG in the transcription factor binding site of the same gene. The missense SNP only has an effect on the function of the gene if the gene is transcribed and the transcription is influenced by the CpG.

To investigate this kind of relationship between CpGs and SNPs, we tried three different approaches.

- 1) Identify a GWAS SNP and a EWAS CpG pair located within 1Mbp (30 pairs identified).
- 2) Identify a GWAS SNP and a EWAS CpG pair located within 10Mbp (192 pairs identified).
- 3) Identify a GWAS SNP and a EWAS CpG pair annotated to the same gene. Here SNPs were annotated to a gene if it is located within the transcript boundary of a protein-coding gene or a nearest gene if it is located outside of genes. Using the Illumina annotation file, a CpG was annotated to a gene if it is located within 1500 base pairs of the transcription start site, 5'-UTR, gene body, or 3'-UTR (see above and method section). Intergenic CpGs were not compared for this 3rd approach.

We only included the result from 3) based on gene annotation in the manuscript previously. In the revised manuscript, we added a new section “Overlap with prior related genome wide association studies” in Method, the following sentence “We identified 30 CpG-SNP pairs within 1Mbp and 192 pairs within 10Mbp (Supplementary Tables 9 and 10).” in Results, and included the results of 1) and 2) as **Supplementary Tables 9 and 10**, respectively.

- It would be informative to mention on which populations (race/ethnicity) previous EWAS and GWAS were performed.

The previous EWAS and GWAS includes non-Hispanic white, non-Hispanic black, Hispanic, and/or Indian Asians. This information was added as **Supplementary Table 11** in the revised manuscript.

- A total of 6 genes had both CpGs and SNPs significantly associated with a blood

lipid'. Instead of only describing the overlap at the gene level, it would be informative to describe the number of CpGs from the EWAS that map to GWAS loci.

As described in the response to your previous comment, we also compared CpGs and GWAS SNPs and

- 1) Identified 30 GWAS SNP and EWAS CpG pairs located within 1Mbp.
- 2) Identified 192 GWAS SNP and EWAS CpG pairs located within 10Mbp.

The results of 1) and 2) were added as **Supplementary Tables 9 and 10**, respectively, in the revised manuscript.

- 'Many of the CpGs identified in our lipid EWAS have been previously identified in EWAS of BMI'. >Please report the exact number of CpGs (or %).

In total, 33 (17%) out of 192 CpGs identified in our race/ethnic stratified analysis of model 4 were previously reported in EWAS of BMI (a total number of 187 CpGs from BMI EWAS in Nature 2017). Also the following sentence "About one half of the genes identified in our BMI adjusted lipid EWAS have also been implicated in previously published BMI EWAS" was revised to "About one half of the genes (33/192=17% of CpGs) identified in our BMI adjusted lipid EWAS have also been implicated in previously published BMI EWAS".

- 'Among EA, 34 out of 106 genes have been identified in EWAS of other phenotypes'. > Why is this only described for Europeans?

Thanks for the comment. In the original manuscript, we made a comparison focusing on Europeans. In the revised manuscript, we added the following sentences to describe the results for African and Hispanic populations: "There were 14 out of 32 genes (14 out of 54 CpGs) and 2 out of 6 genes (4 out of 8 CpGs) have been identified in EWAS of other phenotypes for AA and Hisp, respectively." and updated the **Figure 3** accordingly. The CRP EWAS results were also added as other EWAS and the number was updated from 34 to 37 for European.

- The authors only discuss BMI as an explanation for attenuation of effect sizes in model 2 and 4, and only describe the overlap with CpG sites from previous BMI EWA studies, while model 2 also corrects for smoking. What is the overlap with smoking-associated CpG sites and to what extent is the association between lipids and methylation possibly driven by smoking?

To clarify once more, all 4 models (not only models 2 and 4 but also models 1 and 3) were adjusted for smoking (Please also check out our response to your previous comment regarding smoking).

We checked the overlap with smoking associated CpG sites identified by Joehanes R et al. [Joehanes R et al. Epigenetic Signatures of Cigarette Smoking. *Circ Cardiovasc Genet.* 2016 Oct;9(5):436-447]. Out of 2623 CpGs associated with current smoking, 14 CpGs overlap with our lipid EWAS results (238 CpGs identified in model4). Out of 158 CpGs associated

with former smoking, 1 CpG overlap with our lipid EWAS results (238 CpGs identified in model4).

Given that our all 4 models adjusted for smoking, we were not able to estimate to what extent the association between lipids and methylation are driven by smoking.

In the revised manuscript, we added the comparison results of smoking EWAS and our lipid EWAS in **Figure 3**.

- It is unclear why the authors chose to use a random effects meta-analysis for the EWAS results, and a fixed effects meta-analysis for the mQTL results.

For EWAS, we reported both fixed and random effects meta-analysis results in **Supplementary Table 2**. For EWAS, there were 15 cohorts with three race/ethnic groups and the heterogeneity (I² and Q values also reported in the table) was significant for some of the CpGs hence we focused on random effects meta-analysis. On the other hand, mQTL was performed in 1-4 cohorts of European populations (depending on the availability of a SNP) so we reported fixed effect meta-analysis results, however, we also reported random effects meta-analysis results and heterogeneity measure.

- Note that the term mQTL refers to the SNP, not the CpG. In the manuscript, mQTL is currently used to refer to CpGs at least once.

We are sorry for the confusion. We used the term mQTL to refer a SNP which is associated with the methylation level of a CpG. We made following modifications in the revised manuscript:

“We found 7 out of our 11 CpGs in EA to also be listed as mQTLs in the mQTL DB” -> “We found 7 out of our 11 CpGs in EA to also be listed to have at least one mQTLs in the mQTL DB”

- “Out of the 190 significant mQTLs common to both our study and the mQTL DB, 51 (27%) were found to be mQTLs in datasets spanning across the life course in the mQTL DB including birth, childhood, adolescence, pregnancy, and middle age”. > It is unclear to me if 190 refers to SNPs, CpGs, or SNP-CpG pairs.

As stated above, we used the term mQTL to refer a SNP which is associated with the methylation level of a CpG. Here we meant that there were 190 SNP-CpG pairs. In the revised manuscript we made the following changes to the sentence: “the 190 significant mQTLs” -> “the 190 significant mQTLs (SNP-CpG pairs)”.

- The sample size (total number of participants) of the mQTL analysis, expression analysis and Mendelian Randomization Analysis should be reported in the results section.

In the revised manuscript, we have added the sample size information in the “Methylation quantitative trait loci (mQTL) analysis” section of Results: “Five out of the 15 cohorts

provided genetic data for this analysis including ARIC (N_AA=1,717), GOLDN (N_EA=713), KORA (N_EA=1,379), WHI-BA23 (N_EA=790, N_AA=540, and N_HISP=324), and WHI-EMPC (N_EA=494, N_AA=424, and N_HISP=221).”

In the “Expression quantitative trait methylation (eQTM) analysis” section of Results: “The association between DNA methylation and gene expression were investigated in the Framingham Heart Study (N=4,278 including 2,726 offspring cohort participants and 1,552 third generation cohort participants).”

In the “Mendelian randomization approach” of Results: “We found the GRSs to be significantly associated with their respective lipid levels in the 4 cohorts (GOLDN (N_EA=713), KORA (N_EA=1,379), WHI-BA23 (N_EA=790), and WHI-EMPC (N_EA=494) participating in the MR follow-up analysis.”

- The results section does not mention on which populations (EA, AA or HISP) the expression and Mendelian Randomization analyses are performed.

The expression and Mendelian Randomization analyses were performed in EA. In the revised manuscript, we add the population information in the “Expression quantitative trait methylation (eQTM) analysis section of Results: “The association between DNA methylation and gene expression were investigated in the Framingham Heart Study (N_EA=4,278 including 2,726 offspring cohort participants and 1,552 third generation cohort participants).”

In the “Mendelian randomization approach” section of Results: “We explored the causal relationships between methylation and blood lipid levels for the 30 CpGs in EA (Table 2) using a bi-directional Mendelian Randomization (MR) study design.”

- It would be informative to report the variance in lipid levels explained by the GRSs.

Unfortunately, we did not collect this information from each cohort and are not in a position to easily secure this information. However, we are reassured by the very low p values of the meta-analysis of each of three GRSs we calculated. Also, given the lead SNPs included in the GRS were selected from Teslovich et al. 2010 (PMID: 20686565) and GLGC/Willer et al. 2013 (PMID: 24097068), we are quite confident the GRS each explains approx. 7.5-10.5% of the variance of each of the lipid values based on the performance of similar GRS in an independent cohort, the Million Veteran Program (PMID: 30275531).

- The overlap of SNPs that are included in the GRS for the different lipids, and mQTLs should be discussed. Related to this point, pleiotropy should be discussed as a possible limitation of MR analyses.

The SNPs that are included in the GRS are listed in **Supplementary Table 6**. In European population, six SNPs included in the GRS for HDL (46 SNPs) and LDL (35 SNPs) overlap (rs2954029, rs174546, rs964184, rs4420638, rs9987289, rs3764261); Thirteen SNPs included in the GRS for HDL(46 SNPs) and TG (31 SNPs) overlap (rs4846914, rs2954029, rs4765127, rs174546, rs6065906, rs1042034, rs12678919, rs2972146, rs964184, rs3764261, rs1532085, rs17145738, rs11613352); seven SNPs included in the GRS for LDL (35 SNPs) and TG (31 SNPs) overlap (rs2131925, rs6882076, rs2954029, rs174546, rs964184,

rs10401969, rs3764261); and four SNPs included in the GRS for HDL, LDL, and TG overlap (rs2954029, rs964184, rs174546, rs3764261).

The SNPs in GRS (**Supplementary Table 6**) do not overlap with the mQTL SNPs (**Supplementary Table 3**).

The lipid variables, HDL, LDL, and TG, are closely related to each other and they share genetic backgrounds involved in lipid metabolism. Therefore, we could not rule out biases due to pleiotropy in our MR analysis. In the discussion section of the revised manuscript, we added pleiotropy as one of the limitations of our MR analyses: “In addition, the known shared genetic background of HDL, LDL, and TG introduces the possibility of biases due to pleiotropy in our MR analysis.”

- ‘We implemented inverse-weighted MR method and MR-egger when >2 mQTLs were available for a given CpG.’ > For how many CpGs was this the case? (this should be clarified in the text)

As shown in the sheet of “CpG->TG” of Supplementary Table 7, four out of seven CpGs have one mQTL and the other three CpGs have >2 mQTLs. In the revised manuscript, the information was added: “We implemented inverse-weighted MR method and MR-egger when >2 mQTLs were available for a given CpG (3 CpGs out of 7).”

- It is unclear to me why the 2 directions of causation were tested using different methods (lipids to methylation uses a GRS approach, without consideration of pleiotropy it seems, while methylation to lipids is tested using MR-egger, which is based on summary statistics and does take pleiotropy into account).

The method we selected to test causality was based on the type of instrument available to us.

- 1) From lipids to methylation, there were several large-scale lipid GWAS we used to identify instruments that included a list of hundreds of potential SNPs that could be used. However, it was challenging to identify individual SNPs that were significantly associated with lipid levels in all the cohorts we analyzed in this study in large part because the size of these cohorts was relatively small in relation to the GWAS. However, the GRS calculated from all SNPs were significantly associated with lipid levels in all the cohorts. Under these circumstances, we were concerned that implementing an MR-egger method might introduce a bias from weak instruments. Therefore, we decided to use the GRS based approach.
- 2) From methylation to lipids, we used mQTL results to identify instruments. When mQTLs exist for a CpG, the association signals are individually very strong and thus we were less concerned about weak instruments. Furthermore, the SNPs identified through mQTLs were generally located in close proximity to each other with were often in high LD making a GRS approach less feasible.

- The authors might want to consider adding Manhattan plots from the other groups to figure 1 (it currently only shows results from the analyses in Europeans), because the trans-ethnic component is a major asset of this study.

In the revised manuscript, we revised the **Figure 1** to have Manhattan plots of EA, AA and HA.

- In figure 2, it is unclear to me why HISP and AA are only compared to EA, and not to each other (AA versus HISP).

Our primary reason to only compare HISP and AA to EA was the smaller sample sizes of both HISP and AA leading to a smaller number of significant findings that could be used in correlation and regression analyses. Nevertheless, we now include all 3 comparisons (EA vs AA, EA vs HISP, AA vs HISP) in our revised **Figure 2**.

- The Venn diagrams are informative (figure 3). Would it be useful to include similar Venn diagrams to illustrate the overlap of findings across: European, African, Hispanic, and the trans-ethnic meta-analysis? Perhaps in that case it is more practical to show numbers of CpGs instead of gene names.

We added **Supplementary Figure 5** with the number of significant CpGs information in the revised version of the manuscript.

- Which standard deviation is shown in table 2?

We apologize for the typo. This is standard error of the beta. We corrected this in the revised manuscript.

- Supplemental tables could be organized better. Some have multiple tabs or multiple tables on one tab, and there are no legends. For example, in supplemental table 2, it is unclear what exactly is the purpose of the two sub-tables on the first tab and in table S3, what does TE.fixed and num_studies mean?

In the revised version, the supplementary table has been reorganized to have one table in each sheet with description of each column and the first sheet with the summarized table with hyperlinks to all the other sheets.

Discussion

- How do the authors interpret the weak correlation of effect sizes across ethnicities for LDL compared to the relatively strong correlation of effect sizes across ethnicities for HDL and triglycerides?

A similar concern was brought up by first reviewer. Thus, we refer this reviewer to our response to first reviewer's question "Could to authors explain further why there was not a high correlation between LDL effect sizes? (pg 160)". In brief, the correlations/regressions may have been lower/negative because the substantially smaller number of LDL findings made these analyses much less reliable. We also explored the effect of naturally log transforming LDL given such transformation was applied up front to HDL and TG. We observed modest correlations when considering a larger number of CpGs with relaxed genome wide significance as well as naturally transformed LDL variable. To help readers,

we added the beta comparison plots for 122 CpGs ($P < 1.09 \times 10^{-5}$, with and without natural log transformation) as **Supplementary Figure 4**.

- I think that ‘concordance of effects’ should be ‘concordance of the direction of effects’.

We have corrected this in the revised version of the manuscript.

- There seems to be a discrepancy between the number of CpGs that are associated with gene expression levels in the results section (7) and the discussion section (5).

We apologize for the typo and appreciate the careful review of our manuscript by the reviewer. There are indeed 7 CpGs associated with gene expression levels. We corrected this in the revised version of the manuscript.

- “Third, we found the methylation status of CpGs in CPT1A, a gene that initiates the oxidation of long-chain fatty acids, to be influenced by blood levels of TG through our MR analysis consistent with findings from a previous EWAS of this locus with blood pressure” This sentence is unclear to me. Did the EWAS of blood pressure also find a causal effect of TG on methylation at this locus? Or a causal effect of blood pressure?

The EWAS of blood pressure found that blood pressure influences methylation at cg00574958 located in *CPT1A* (Richard MA et al., *Am J Hum Genet.* 2017 Dec 7;101(6):888-902). In the revised manuscript, we revised the sentence to be the following: “Third, we found the methylation status of CpGs in CPT1A, a gene that initiates the oxidation of long-chain fatty acids, to be influenced by blood levels of TG through our MR analysis. The same CpG (cg00574958) in CPT1A was also found to be influenced by blood pressure levels in another EWAS [ref 24].”

- How do the results of the MR analyses compare to findings from the previous MR study on lipids and methylation by Dekkers et al (reference 14)?

The DNA methylation levels of two CpGs (cg00574958 and cg17058475) in CPT1A decreased by blood levels of TG in both of our study and Dekkers’s study.

- Is it possible to add a reference for the sentence “While circulating leucocytes are likely to exert at least partial direct control over blood lipid levels”?

We provide 2 reference to support this statement including PMID: 20844574 and 28764798 (References 50 and 51).

Methods

- The description of correction for family structure is very brief. I expected to find more details about this in the supplemental methods (i.e. what type of family

relationships are present in which cohort, and if multiple degrees of relatedness are present, are these modeled by inclusion of multiple random effects?)

Four cohorts (FHS, GOLDN, TwinsUK, and GENOA) were family-based cohorts (Footnote of **Table 1**). Detailed information of each of cohort is included in **Supplementary Methods** and related references. For instance, “Framingham Heart Study (FHS): The Framingham Heart Study Offspring cohort (FHS-Offspring) was initially recruited in 1971 and included 5,124 offspring of the FHS Original cohort. From 2002 to 2005, the adult children (third generation cohort, N=4,095) of the offspring cohort participants were recruited and examined (FHS-3rd Gen). For each family based cohort, relatedness was indeed modeled appropriately by analysts who were acutely aware of the cohorts ascertainment scheme as has been done for other EWAS published using these cohorts.

- Are the same white blood cell proportions included in all analyses (i.e. EWAS, mQTL, eQTM, MR?). From the description of the EWAS analyses, it seems that all cohorts used Houseman estimates, while from the description of the mQTL analyses, it seems that some cohorts used measured cell counts while other used Houseman estimates.

Yes, the same white blood cell proportions were included in EWAS, mQTL, and eQTL but not included in MR. All cohorts except GOLDN used cell proportions estimated from Houseman correction. GOLDN used CD4+ (**Supplementary Table 1**).

- In formula for the GRS, it seems odd to me that the allele dosages were not multiplied by the weights for each SNP from the GWAS. Is this an error?

Thanks for the comment. This is not an error. When we construct a GRS, allele dosages are often multiplied by weights (usually beta estimates from previous GWAS). However, the beta estimates also fluctuate depending on study population and race/ethnicity. Since we only utilized the GRS as an instrument to randomize the exposure, we thought the weights may not be critical and it would be reasonable to assign equal weights for each SNP.

Reviewers' comments:

Reviewer #1 (Remarks to the Author):

Please see attachment

Reviewer #2 (Remarks to the Author):

I thank the authors kindly for their answers to my questions. All my comments have been addressed and I now find the manuscript very clear to read. I have no further comments.

Reviewer #1 Attachment

- Jhun et al. have made an excellent effort in answering the reviewers' concerns including my specific points. However, their response to the query regarding how the overlap with prior GWAS was performed could be much more nuanced. They have chosen arbitrary cut-offs and discuss missense variants - which are the minority of GWAS SNP findings. Significant biases will be at play regarding the array design in relation to probe density per gene, as well as gene length for the gene annotation results. The use of non-biological uniform regional arbitrary cut-offs of 1Mb and 10Mb cut-offs would also need to be justified and assessed - potentially by permutation.
- Also, their overlap with BMI EWAS should not just be with the one EWAS cited as other large-scale BMI-EWAS have also been performed - particularly Demerath et al. [PMID: 25935004] as included African Americans.

Reviewer 1's additional comment

- Jhun et al. have made an excellent effort in answering the reviewers' concerns including my specific points. However, their response to the query regarding how the overlap with prior GWAS was performed could be much more nuanced. They have chosen arbitrary cut-offs and discuss missense variants - which are the minority of GWAS SNP findings. Significant biases will be at play regarding the array design in relation to probe density per gene, as well as gene length for the gene annotation results. The use of non-biological uniform regional arbitrary cut-offs of 1Mb and 10Mb cut-offs would also need to be justified and assessed - potentially by permutation.

We are unaware of biases in the probe density of the genes that related to the collective set of lipid GWAS loci discovered by the Global Lipids Genetics Consortium (GLGC) between 2010-2013. We note that the GLGC discoveries were reported after the Illumina 450k was designed. The description of the array can be found at https://www.illumina.com/content/dam/illumina-marketing/documents/products/datasheets/datasheet_humanmethylation450.pdf. The section under "comprehensive genome-wide coverage" states that 99% of RESeq genes are covered. The additional "high-value content selected with guidance of methylation experts" described at the end of this section does not suggest to us that the high value content is specifically related to the ~100 genomic regions discovered by GLGC to be associated with lipids in 2010-2013. Furthermore, our CpG-lipid pairs are only declared significant after taking into consideration a very conservative multiple testing p value threshold (Bonferroni).

We would like to clarify that the missense was only one example mentioned to help illustrate how we could link CpG and SNP to the same gene. As our manuscript describes, we actually considered a total of 60 combinations = 10 for a SNP (missense, 5'UTR, insertion, noncoding exon, splice donor, regulatory region, stop, synonymous, intron, and 3'UTR) × 6 for a CpG (200-1500bp of transcription start site, 0-200bp of transcription start site, 5'UTR, 1st exon, gene body, and 3'UTR) when linking a CpG to a SNP through a given gene.

The 1M and 10Mb cutoffs may appear somewhat arbitrary given that at least some significantly associated SNP-CpG pairs may be located further apart on the same chromosome or sometimes even on different chromosomes (trans-methylation quantitative loci). However, such large distances between SNP methylation QTL and the respective CpGs are in the minority. In fact, Liu et al (PMID 24656863) found that 93% of 98,658 significant vCpG-SNP associations in blood using the methylation 450k array were within 5Mb distance of each other confirming observations of others that a majority of genetically influenced methylation is controlled through cis-regulation. Thus, a distance of 10 mb is expected to identify a very large majority of cis-meQTL pairs within a specific tissue. Incidentally, we compared all of our lipid GWAS SNP-lipid EWAS CpG pairs to the 98,658 SNP-CpG pairs identified by Liu et al. and found that none of the Lipid GWAS SNP-EWAS CpG pairs overlapped with the previously identified SNP-CpG pairs. Of course, such methylation QTL links may exist in other relevant tissues for lipids, a link could still be present through independent regulatory mechanisms on the same lipid gene.

- Also, their overlap with BMI EWAS should not just be with the one EWAS cited as other large-scale BMI-EWAS have also been performed – particularly Demerath et al. [PMID: 25935004] as included African Americans.

All the cohort (ARIC, FHS, and GOLDN) participated in the BMI study by Demerath et al. participated in our study. The discovery population (ARIC) was African American population while the replication populations (FHS and GOLDN) were European American populations. We updated Figure 3 to show the overlap with the results of the BMI EWAS study by Demerath et al.